# A new efficiency metric for the spatial evaluation and inter-comparison of climate and geoscientific model output

Andreas Karpasitis[1], Panos Hadjinicolaou[1], and George Zittis[1]

[1]Climate and Atmosphere Research Center (CARE-C), The Cyprus Institute

**Correspondence:** Andreas Karpasitis (a.karpasitis@cyi.ac.cy)

**Abstract.** Developing and evaluating spatial efficiency metrics is essential for assessing how well climate or other models of the Earth's system reproduce the observed patterns of variables like precipitation, temperature, atmospheric pollutants, and other environmental data presented in a gridded format. In this study, we propose a new metric, the Modified Spatial Efficiency (MSPAEF), designed to overcome limitations identified in existing metrics, such as the Spatial Efficiency (SPAEF), the Wasserstein Spatial Efficiency (WSPAEF), or the Spatial Pattern Efficiency metric ($E_{sp}$). The performance of MSPAEF is systematically compared to these metrics across a range of synthetic data scenarios characterized by varying spatial correlation coefficients, biases, and standard deviation ratios. Results demonstrate that MSPAEF consistently provides robust and intuitive performance, accurately capturing spatial patterns under diverse conditions. Additionally, two realistic but synthetic case studies are presented to further evaluate the practical applicability of the metrics. In both examples, MSPAEF delivers results that align with intuitive expectations, while the other metrics exhibit limitations in identifying specific features in at least one case. Finally, as a real-world application, we rank global Coupled Model Intercomparison Project phase 6 (CMIP6) model data according to their skill in representing precipitation and temperature using the four different metrics. This application highlights that the MSPAEF rankings are most similar with $E_{sp}$ with a normalized absolute ranking difference of 2.8 for precipitation, and 3.8 for temperature. These findings highlight the added value of the MSPAEF metric in evaluating spatial distributions and its potential to be used in climate or other environmental model evaluation or inter-comparison exercises.

## 1 Introduction

An accurate and comprehensive evaluation of climate models is crucial for understanding their limitations, enhancing the representation of processes, and distilling the most credible information on regional climate changes, impacts, and risks (Eyring et al., 2019). Climate models and their output are essential tools for scientists, practitioners and policymakers. However, before any application, it is important to use robust validation metrics to assess their performance and accuracy (Wagener et al., 2022). Traditional evaluation methods often emphasize temporal correlations or mean biases, while they often overlook crucial spatial patterns that significantly affect the model performance assessment. Capturing these spatial distributions is particularly important for environmental variables like precipitation and temperature, as the spatial variation greatly influences regional processes and subsequent applications, such as impact assessments (Citrini et al., 2024; Ramirez-Villegas et al., 2013).

In this paper, the term metric is used in a broad sense to refer to all indicators, statistics or distance measures that take as input two datasets and output a value that quantifies their relative performance or similarity. This includes, but is not limited to, quantities that satisfy the formal definition of a metric.

Many studies focus their evaluation of climate models on more simple metrics such as the Normalized Mean Square Error (NMSE) (Simpson et al., 2020), Bias (Mehran et al., 2014; Su et al., 2013), the Root Mean Square Error (RMSE) (Sillmann et al., 2013; Kamworapan and Surussavadee, 2019; Srivastava et al., 2020; Li et al., 2021; Nishant et al., 2022; Kim et al., 2020b), the Mean Absolute Error or Deviation (MAE or MAD) (Zittis and Hadjinicolaou, 2017; Lovino et al., 2018), the Relative Error (RE) (Kim et al., 2020a), or the Skill Score (Srivastava et al., 2020). For instance, for the evaluation and inter-comparison of downscaling approaches for climate models, Maraun et al. (2015) used a variety of metrics such as the Relative Error (RE) for the spatial evaluation of different downscaling methodologies, while for the temporal evaluation, performance indicators such as the mean square error (MSE), the correlation coefficient, the bias and Relative Error were used. A different concept was introduced by Brands et al. (2011) who evaluated global General Circulation Models (GCMs) using the overlap of the Probability Density function of the time series for each grid box between the model and the observational dataset. The Kolmogorov-Smirnov test (KS test) is another metric that was used for the evaluation of models contributing to the Coupled Model Inter-comparison Project (CMIP) (Brands et al., 2013). Kotlarski et al. (2014) evaluated a large ensemble of Regional Climate Models (RCM) contributing to the EURO-CORDEX initiative using a multitude of metrics, including spatial efficiency ones. This selection includes the spatial pattern correlation coefficient of time-averaged values (PACO), and the ratio of the spatial standard deviation (RSV) of time-averaged values between the model and the reference dataset. The Pearson Correlation Coefficient can also be used in the time series of seasonal means of a variable of each grid-box of the data, to assess how well a climate model represents the local location representativeness (Maraun and Widmann, 2015). While these metrics can be effective for certain applications, their performance can vary significantly based on the statistical and spatial characteristics of the data. For instance, the Pearson correlation coefficient is useful for capturing the linear relationship between two datasets; however, it does not account for systematic biases in the mean or differences in variance (i.e., scale differences). Similarly, other metrics like RMSE and MAE are sensitive to both magnitude and distribution of errors, but they may be disproportionately affected by a large bias in the mean, which can lead to a misrepresentation of spatial patterns. In contrast, the Kolmogorov-Smirnov test compares the underlying distributions of two datasets but lacks spatial context, which is often crucial in geoscientific modeling. These distinct limitations underscore the challenges of using traditional metrics on variables with uneven spatiotemporal distributions, like precipitation, which is often sparsely distributed in space. They also complicate comparisons between models that possess differing statistical properties, such as varying means or levels of variation.

Compound metrics that integrate several simpler approaches have been developed to assess the performance of geoscientific models more accurately. A commonly used goodness-of-fit indicator is the Kling-Gupta Efficiency (KGE), which is a robust compound metric, commonly used in hydrological sciences for comparing simulations to observations (Gupta et al., 2009; Deepthi and Sivakumar, 2022; Zittis et al., 2017). The SPAtial EFficiency metric (SPAEF) proposed by Demirel et al. (2018b, a) was inspired by KGE (Koch et al., 2018) and has emerged as a promising tool for evaluating hydrological models by

considering three key aspects of spatial pattern accuracy: correlation coefficient, relative variability, and distribution of values in standardized space. This metric has also been applied to climate model evaluation, for the ranking of the performance of precipitation and temperature of different climate models (Lei et al., 2023; Deepthi and Sivakumar, 2022; Ahmed et al., 2019; Verma et al., 2023), as well as for the evaluation of precipitation in reanalyses datasets (Gomis-Cebolla et al., 2023). SPAEF combines various elements, equally weighted, into a single metric, offering a more comprehensive assessment of spatial distribution compared to traditional methods like mean-squared error or correlation alone. While SPAEF can be useful for evaluating climate models, it has some limitations, for example, it is insensitive to the magnitude of biases. Although this feature can be advantageous in certain contexts, an assessment of biases is crucial for a complete evaluation of a model and should be taken into account.

An existing modification and improvement of SPAEF is the Wasserstein SPAEF (WSPAEF) (Gómez et al., 2024). Unlike the original version, WSPAEF is sensitive to model biases. However, its sensitivity is based on the absolute magnitude of the bias, making it inherently dependent on the units of the variable used. This means that the metric's response to bias may vary across different variables or datasets with different scales, which should be considered when interpreting results. Another common limitation shared by both approaches is that two out of their three components—the standard deviation and differences in histogram values for SPAEF and the standard deviation and Wasserstein distance for WSPAEF—primarily describe the overall distribution of the parameter under evaluation rather than its spatial structure. This leaves only one component that evaluates the model variable based on the pattern of the variable, which is the spatial correlation coefficient.

Another indicator that was inspired by SPAEF is the Spatial Pattern Efficiency Metric ($E_{sp}$) (Dembélé et al., 2020). This metric, like SPAEF, is bias-insensitive, and focuses on the spatial distribution of the patterns of a variable. This metric has been used to evaluate the spatial patterns of the hydrological processes in gridded precipitation datasets (Dembele et al., 2020). The SPAEF metric and some of its proposed modifications have been comprehensively evaluated by Yorulmaz et al. (2024).

Here, in response to the need for a comprehensive and multi-faceted evaluation of climate and other earth system models, we propose a new metric. This is based on a modification of SPAEF and is designed to more precisely capture the spatial distribution characteristics of climate model output. This new approach addresses certain limitations of existing metrics by incorporating spatial components that improve sensitivity to the spatial distribution and the relative bias of the variable under evaluation. We apply this new metric to synthetic datasets that imitate statistical properties and possible distributions of two types of climate variables, demonstrating its ability to offer improved insight into the spatial fidelity of model outputs. Through this work, we aim to provide the climate modeling community with a refined tool for assessing spatial distribution accuracy, contributing to more reliable model evaluations, improved climate model development, and, ultimately, more accurate projections. The proposed approach holds significant potential for applications in other scientific fields, including hydrology and environmental sciences.

It is important to stress that the MSPAEF metric is not intended to replace traditional multi-metric evaluation approaches. Instead, it is designed to complement existing measures by providing a balanced indicator that captures both spatial pattern similarity and relative mean bias within a single metric. In this way, MSPAEF serves as a useful additional tool, particularly when a unified summary statistic is desirable.

 **2 Data and Methods**

## 2.1 Metrics

This study focuses on compound metrics specifically designed for the spatial evaluation of climate or other environmental parameters. The four metrics evaluated and inter-compared in this work are presented below.

**SPAEF metric**

The SPAEF (SPAtial EFficiency) metric by Demirel et al. (2018b) is a robust metric that was originally created for hydrological model evaluation, and it is used to characterize the performance of a model regarding the spatial distribution of a variable. It is defined as:

$$SPAEF = 1 - \sqrt{(\alpha - 1)^2 + (\beta - 1)^2 + (\gamma - 1)^2} \tag{1}$$

with:

$$\alpha = \rho(M, O) \quad , \quad \beta = \frac{\left(\frac{\sigma_M}{\mu_M}\right)}{\left(\frac{\sigma_O}{\mu_O}\right)} \quad \text{and} \quad \gamma = \frac{\sum_{j=1}^{n} min(K_j, L_j)}{\sum_{j=1}^{n} K_j} \tag{2}$$

where $\rho$ is the spatial correlation coefficient between the points of the model and the observations, $\sigma$ and $\mu$ are the standard deviation and mean value for each of the model and observation, $K$ and $L$ are histograms with n common bins, of the standardized values (z-score) of the model and observations respectively, and $\gamma$ is the histogram overlap of the standardized values of the two datasets. This metric is bias-insensitive since none of its three terms are affected by present bias, and therefore the final metric conveys only the similarity in the patterns between the model and the observations. The SPAEF metric takes values from $-\infty$ to 1, with 1 indicating an excellent agreement of the model with the observations.

**The Spatial Pattern Efficiency Metric ($E_{sp}$)**

The $E_{sp}$ metric by Dembélé et al. (2020) was designed to be a simple and robust metric for hydrological model evaluation, and it is defined as:

$$E_{sp} = 1 - \sqrt{(r_s - 1)^2 + (\gamma - 1)^2 + (\alpha - 1)^2} \tag{3}$$

with:

$$r_s = 1 - \frac{6\sum_1^n d^2}{n(n^2 - 1)} \quad , \quad \gamma = \frac{\left(\frac{\sigma_M}{\mu_M}\right)}{\left(\frac{\sigma_O}{\mu_O}\right)} \quad \text{and} \quad \alpha = 1 - E_{RMS}(Z_M, Z_O) \tag{4}$$

where $r_s$ is the Spearman rank-order correlation coefficient, $\sigma$ and $\mu$ are the standard deviation and mean value for each of the model and observation, $Z_M$ and $Z_O$ are the standardized values (z-score) of the model and observations respectively, and $E_{RMS}$ is the root mean square error of the standardized values. A benefit of this metric is that it does not require any user-defined parameters, such as the bins of the histograms, and that two of the three components take into account the spatial distribution of the variable. It takes values between $-\infty$ and 1, with 1 indicating excellent performance of the model.

**WSPAEF Metric**

An relatively newly developed modification of the SPAEF is the Wasserstein Spatial Efficiency (WSPAEF) by Gómez et al. (2024) and is defined as:

$$WSPAEF = \sqrt{(\alpha - 1)^2 + (\sigma - 1)^2 + (\phi)^2} \tag{5}$$

with:

$$\alpha = \rho(M,O) \quad , \quad \sigma = \frac{\sigma_M}{\sigma_O} \quad \text{and} \quad \phi = WD = \left(\sum_{i=1}^{n} |X_{(i)} - Y_{(i)}|^p\right)^{\frac{1}{p}} \tag{6}$$

where $\rho$ is the spatial correlation coefficient between the points of the model and the observations, $\sigma$ is the standard deviation for each of the model and observation, WD is the Wasserstein distance of order p=2, with $X_{(i)}$ and $Y_{(i)}$ being the i-th order statistic of the samples of the observations and model, respectively, and n is the total number of samples in each dataset. This means that the values of the two datasets have been arranged in ascending order, and $X_{(i)}$ is the i-th value in the sorted list. WD was calculated using the original values of the two datasets to explicitly account for the bias. This modification of SPAEF is sensitive to bias since the Wasserstein distance term is a measure of the minimum required effort that is needed to change the histogram of one dataset to match the histogram of the other dataset. It takes values from 0 to $\infty$, with values approaching zero indicating excellent agreement of the model with the observations.

**MSPAEF Metric**

According to these definitions, some limitations can be introduced when using the three previous metrics. For instance, SPAEF and WSPAEF only include one component that considers the spatial distribution of the variable's values, which is the correlation coefficient in the space domain, while their other two components take into consideration the overall distribution of the parameter. Furthermore, SPAEF and $E_{sp}$ are bias-insensitive metrics. While this characteristic may be beneficial in certain contexts, it is not ideal for a comprehensive assessment of climate models, where capturing both spatial patterns and biases is essential for a thorough evaluation. In addition, all three of them are scale-dependent, meaning that their values change based on the units of the input variable.

Therefore, we propose a new metric, on the basis of SPAEF that is bias-sensitive, but at the same time scale-independent. This metric conveys information not only about the spatial patterns but also the biases, like the WSPAEF. This is desired because, for a more complete evaluation of the spatial characteristics of a modeled parameter, both the pattern information, and the bias are essential. This metric, like SPAEF and $E_{sp}$, takes values from negative infinity to 1, with 1 indicating a perfect agreement of the model compared to the observations. The proposed modification to the SPAEF is the following:

$$MSPAEF = 1 - \frac{1}{\sqrt{4}}\sqrt{(\alpha - 1)^2 + (\beta)^2 + (\gamma)^2 + (\delta)^2} \tag{7}$$

with:

$$\alpha = \rho(M,O) \quad , \quad \beta = NRMSE \quad , \quad \gamma = \frac{|\overline{M} - \overline{O}|}{IQR_o} \quad , \quad \delta = \sqrt{(\sigma^2 - 1)^2 + \left(\frac{\sigma^2 - 1}{\sigma^2}\right)^2} \tag{8}$$

also:

$$NRMSE = \frac{1}{IQR_O} \sqrt{\frac{\sum_i^n (M_i - O_i)^2}{n}} \quad \text{and} \quad \sigma = \frac{\sigma_M}{\sigma_O} \tag{9}$$

where $\rho$ is the spatial correlation coefficient between the grid cells of the model and the observations, $NRMSE$ is the Root Mean Square Error calculated with the original data and normalized with the inter-quartile range of the observational data ($IQR_O$), while $\sigma_O$ and $\sigma_M$ are the standard deviations of the observations, and the model respectively.

This modification of the SPAEF metric is sensitive to the relative value of bias, while at the same time capturing the spatial pattern differences. Specifically, the correlation coefficient ($\alpha$ term) ensures that the metric reflects the spatial agreement between model output and observations. The normalized RMSE ($\beta$ term) is computed using the interquartile range (IQR) of the observations rather than the standard deviation, as IQR provides a more robust measure of variability (Rousseeuw and Hubert, 2011; Huber and Ronchetti, 2009). This term captures both spatial pattern agreement and bias, behaving differently depending on the magnitude of bias. When bias is small, it emphasizes spatial differences, whereas, in cases of large bias, it is primarily driven by magnitude differences. The relative bias term ($\gamma$ term) is also normalized by the interquartile range of the observations and is introduced to explicitly quantify systematic differences in the mean. Finally, the $\delta$ term accounts for differences in the spread of values by incorporating the standard deviation ratio between the model and observations. This final term comprises two components to ensure symmetry around a value of 1, thereby ensuring the same result regardless of whether the standard deviation ratio or its reciprocal is applied.

An advantage of MSPAEF over SPAEF is that its definition does not rely on user-defined parameters. The MSPAEF metric is formulated without such arbitrary choices, unlike SPAEF, which requires user-specified bins for the histogram overlap term. This makes it more objective and consistent across different datasets, reducing the influence of subjective parameter selection on the results.

A further advantage of normalizing all components by IQR and expressing each term in dimensionless form is that MSPAEF becomes relatively insensitive to preprocessing decisions such as unit conversions, domain rescaling, or masking of small regions. Unlike WSPAEF, whose values are affected by the unit of choice, MSPAEF can produce comparable values under a wide range of analysis settings, which increases reproducibility across studies and domains.

An important motivation for this definition of MSPAEF is to penalize inconsistent model behavior, where a model with poor spatial correlation but a very small bias (or vice versa) can appear artificially good in composite metrics. Since MSPAEF treats each discrepancy in either the spatial patterns or in the mean relative bias as an orthogonal dimension, it prevents a strong performance in one characteristic from masking deficiencies in the other. This provides a more complete evaluation of how each model differs from observations.

Based on its definition, MSPAEF is sensitive to both spatial pattern agreement and mean bias, and this sensitivity can be directly interpreted in the corresponding spatial maps. Each component of the metric can be associated with an observable feature in the maps. The spatial correlation term represents how well the locations and intensities of spatial gradients align between the model and observations. The NRMSE identifies differences in grid-point magnitudes across the domain that are linked to both the location and intensity of gradients, but also to the mean bias. The relative mean bias term quantifies

systematic offsets in the average values of the field. Finally, the variability term reflects how well the model reproduces the amplitude of spatial fluctuations. Together, these components allow changes in the MSPAEF value to be directly linked with spatial mismatches.

## 2.2 Synthetic Data generation

For the development of MSPAEF and the comparison with existing metrics we used a methodology inspired by Gómez et al. (2024). This approach uses synthetic data for both the observations and the model, in order to identify the behavior of the metrics for a variety of predefined combinations of correlation, bias and standard deviation ratio between the two sets of data.

A 10x10 matrix of random data that incorporates some spatial correlation between the different grid points was created using a covariance model. To achieve that, a distance matrix was first created for the specific grid size, which holds the Euclidean distance of all possible pairs of points. Then, the covariance matrix was calculated using this distance matrix and the Mattern covariance function:

$$M_v(r) = \frac{2^{1-v}}{\Gamma(v)} * \frac{r_s^v}{l_s} * K_v(r_s) \tag{10}$$

with:

$$l_s = 1.0 \quad , \quad v = 1.5 \quad , \quad \Gamma(v) = \int_0^\infty t^{v-1}e^{-t}dt \quad and \quad r_s = \sqrt{2*v*r} \tag{11}$$

where $v$ is the smoothing parameter, $r$ is the distance matrix, $r_s$ is the scaled distance matrix, $l_s$ is a scaling factor, $\Gamma(v)$ is the gamma function and $K_v$ is the modified Bessel function of the second kind.

To create the synthetic field with prescribed properties such as a specific standard deviation and mean value, the multivariate normal function was used. This function requires two inputs: a mean vector $\mu$ that specifies the expected value for each point on the grid and a covariance matrix $\Sigma$ encoding the relationships (e.g., variances and covariances) between the values at all pairs of points on the grid. For the mean vector, a vector that holds the target mean values was created. As a covariance matrix input, the following matrix was used in order to achieve the desired standard deviation:

$$C = (\frac{\sigma}{\sigma_M})^2 * M_v(r) \tag{12}$$

with

$$\sigma_M = \sqrt{\frac{Tr(M_v(r))}{d}} \tag{13}$$

where $\sigma$ is the target standard deviation, $M_v(r)$ is the Mattern Covariance function calculated above and d is the number of spatial points. The output of the multivariate normal function represents the spatial distribution of the observational dataset.

### 2.3 Correlated data generation

A second gridded dataset was then created with some correlation, standard deviation ratio and bias target compared to the first grid, to simulate the spatial distribution of the model dataset, using the following equation:

$$y = \lambda_t * \left( \rho_t * (x - \overline{x}) + \sqrt{1 - \rho_t^2} * A * \sigma_x \right) + \overline{x} + \delta_t \tag{14}$$

where $\lambda_t$ is the standard deviation ratio target, $\rho_t$ is the correlation coefficient target, $\delta_t$ is the bias target, $x$ is the original matrix, $\overline{x}$ is the mean value of the original matrix $x$, $\sigma_x$ is the standard deviation of the original matrix $x$ and $A$ is a matrix of the same dimensions as x, filled with random numbers drawn from the normal distribution.

### 2.4 Skewed data generation

The above data are meant to represent climate variables that have a normal distribution since the random data used follow a normal distribution. Nevertheless, numerous climate and other geoscientific model output variables do not follow a normal distribution, but instead exhibit skewed or exponential distributions, as is the case with daily precipitation (Ensor and Robeson, 2008). To simulate skewed distributions, the normally distributed observational values were transformed using exponentiation. A corresponding model dataset was generated by first establishing a specific correlation and standard deviation ratio between the underlying normally distributed values, and then introducing the desired bias after the exponentiation transformation was applied to both datasets. A limitation of this approach is that even though the target correlation coefficient is generally well-preserved through the exponentiation, the target standard deviation ratio and bias are not directly translated to the skewed data. Nevertheless, relative changes in the target standard deviation ratio and bias are reflected in the resulting skewed data.

Using the original observational grid defined above, the skewed observations and model grids are defined as:

$$\begin{aligned} x_s &= e^{x_{rm}} \\ y_s &= e^{\lambda_t * \left( \rho_t * (x_{rm}) + \sqrt{1 - \rho_t^2} * A * \sigma_x \right)} + \delta_t \end{aligned} \tag{15}$$

with:

$$x_{rm} = x - \overline{x} \tag{16}$$

where $x$ is the original matrix, $x_{rm}$ is the original matrix after removing the mean, and the other terms are the same as defined in Eq. 14.

Although the data created this way follow a highly positively skewed distribution, their shape closely resembles an exponential distribution when visualized with an insufficient number of histogram bins, due to the generation process, as seen in Fig. 1.

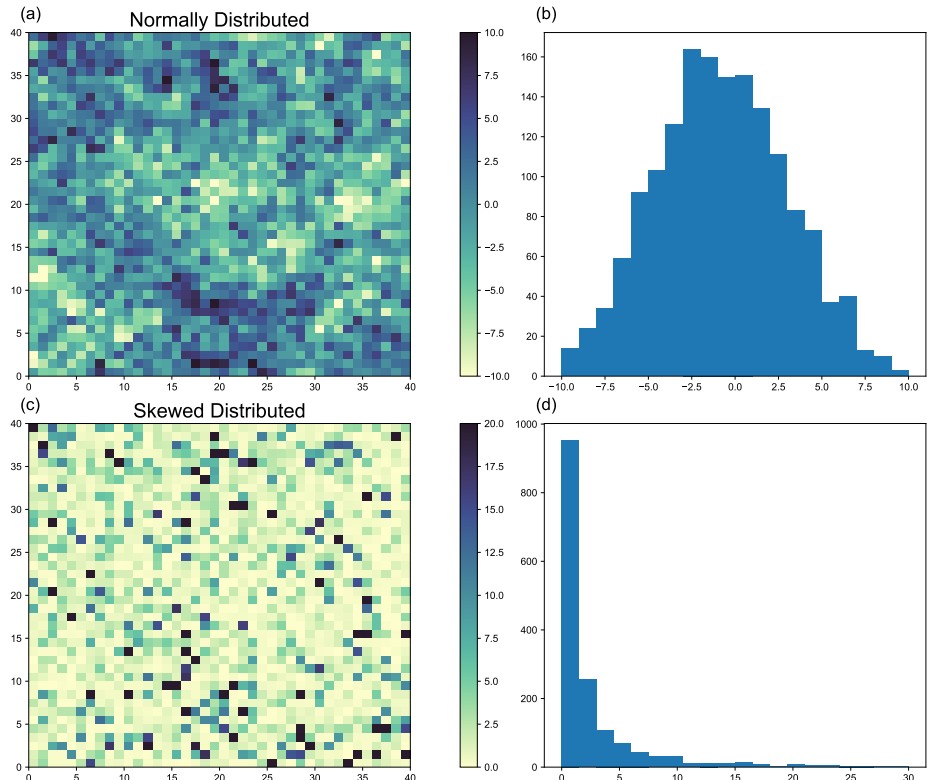

**Figure 1.** Examples of synthetic data generation. Panels (a) and (b) show the spatial distribution and probability density function (PDF), respectively, of synthetic data generated with a normal distribution. Panels (c) and (d) show the corresponding spatial distribution and PDF for skewed data.

The four metrics were calculated for many combinations of bias, standard deviation ratio and correlation, for both the normal and skewed distribution cases. For each combination of the aforementioned parameters, and for both the normally and skewed distributed cases, the procedure was repeated 200 times, and the median value of each metric was used to ensure convergence of the metrics. While the synthetic data are generated to match specific target values of correlation, bias, and standard deviation ratio between them, the stochastic nature of the process means these targets are only approximately achieved. Additionally, non-trivial spatial variations may still occur across realizations, which can affect different metrics in distinct ways. The repetition and the use of the median metric values help reduce the influence of these variations and provide a more robust estimate of each metric's behavior under the intended conditions. To facilitate an easier comparison of the performance of the metrics, these were modified slightly so that they take values equal to or greater than zero, with zero indicating perfect agreement with the observations (see Appendix A).

In these adjusted conditions, where zero indicates a perfect match with the observations, a well-behaved metric is expected to show decreasing values as the correlation coefficient increases and the bias decreases. In the $\lambda$-$\delta$-$\rho$ plots, this is reflected as the curves shift at lower coordinates values as one moves towards the right and upward parts of the subplots, as illustrated by

the purple curves in Fig. 2. Additionally, the lowest metric value for any combination of correlation and bias is anticipated at a standard deviation ratio of 1. This is reflected in the curve minimum being at a standard deviation ratio of 1 for each subplot, with metric values increasing as the standard deviation ratio deviates from 1.

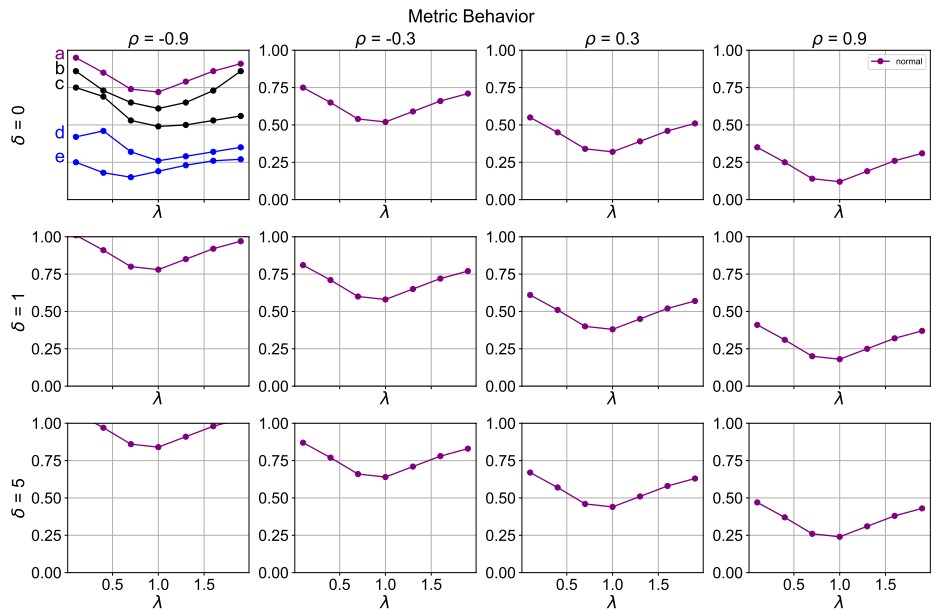

**Figure 2.** Examples of metric behaviour curves for a range of correlation and bias values. Black and purple curves indicate a well-behaved metric, while the blue curves indicate a poorly behaving metric. In the figure, $\lambda$ is the standard deviation ratio target, $\delta$ is the bias target, and $\rho$ is the correlation target between the model and the observations.

In the examples of the top left subplot of Fig. 2, the black and purple curves indicate well-behaving metrics, since for all of them, the minimum values are found at $\lambda$ of 1, and the values increase monotonically as the standard deviation ratio deviates from 1, even though some of the curves are not perfectly symmetric. The blue curves indicate poorly behaving metrics because, for curve e, the minimum value is not found at a standard deviation ratio of 1, while for curve d, the curve does not monotonically increase to the left side of the minimum."

**2.5   Real climate data**

The sixth phase of the Coupled Model Intercomparison Project (CMIP6) is the latest generation of CMIP, which is designed to improve understanding of past, present, and future climate change driven by greenhouse gas emissions. Numerous research institutions worldwide contributed their climate models, following standardized protocols for radiative forcing and data formatting. CMIP6 models have generally shown improved performance in simulating various climate fields compared to previous

CMIP generations (Li et al., 2020). They also exhibit reduced biases in tropical regions, such as the double ITCZ bias (Tian and Dong, 2020). In this study, 33 CMIP6 models were evaluated against the ERA5 reanalysis dataset (Hersbach et al., 2020).

For each model, the ensemble mean of all available variants was used, for precipitation and 2-meter temperature during the historical period from 1981 to 2010. The CMIP6 models and the ERA5 dataset were remapped to 1-degree resolution using linear interpolation for the temperature variable and first-order conservative interpolation for the precipitation variable. For

precipitation, the mean annual total in millimeters was calculated over the period, while for temperature, the mean annual value in Kelvin was computed.

To compute MSPAEF or any other spatial metrics, the model and observational datasets need to be placed into a common grid. Regridding data can generally affect the values of metrics that rely on grid-point values. However, the relative performance rankings among models are expected to remain largely unchanged because all models undergo the same regridding procedure

to the same target grid. Therefore, any smoothing or distortion that occurs due to the interpolation is expected to have a similar and small effect across the datasets, provided that the regridding does not involve extreme changes to resolution.

## 3  Results and Discussion

In Sect. 3.1 and 3.2, the MSPAEF, SPAEF and $E_{sp}$ metrics were modified slightly so that their best performance is indicated at the zero value, like in the WSPAEF metric, for easier inter-comparison of their performance. Contrariwise, in Sect. 3.3, all

metrics were applied in their original form.

### 3.1  Metrics Behavior

For each of the four metrics examined, we calculated its value for a range of values of spatial correlation coefficient, bias and standard deviation ratio between the observational data and the model. The correlation coefficient, bias, and standard deviation ratio were each sampled at discrete intervals within the following ranges: correlation (-0.9 to 0.9), bias (0 to 3), and standard

deviation ratio (0.3 to 1.8). The following plots show the behavior of the four metrics following the logic presented in Fig. 2, using a mean of 10 and a standard deviation of 1 for the observational data.

For the SPAEF metric (Fig. 3), the values generally decrease with increasing correlation, which is the expected behavior of a well-defined spatial efficiency metric. In addition, a good behavior is observed for each curve in the cases where there is no bias present, as the minimum of the curves in these cases is found at standard deviation ratios of 1. Nevertheless, as

the bias increases ($\delta = 1.0, 3.0$), the minimum of the curves generally shifts to values of a standard deviation ratio greater than 1. Specifically, for the skewed distributed variable, it moves for both cases where there is a bias present, while for the normally distributed variable, the shift of the minimum to standard deviation ratios values greater than 1 is observed only for the cases of a large bias ($\delta = 3.0$), with the shift of the normally distributed cases being more perceivable with a high correlation coefficient ($\rho = 0.9$). This indicates that the metric does not perform as intended in larger biases, especially in the case of a

high correlation coefficient, where this shift of the minimum is more evident. In addition, the increasing bias does not directly affect the magnitude of the SPAEF values, highlighting that this metric is bias-insensitive.

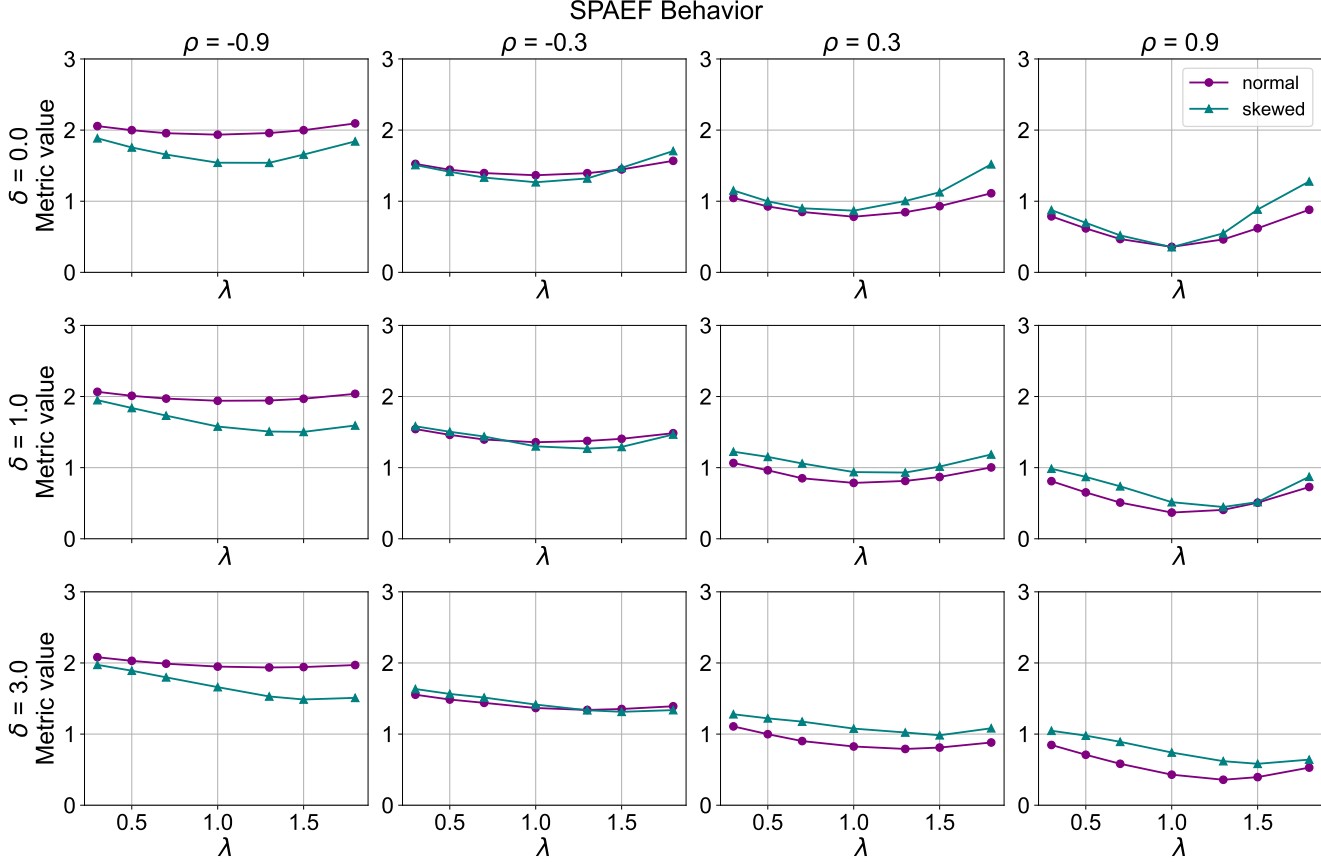

**Figure 3.** SPAEF metric behavior ($\lambda$ is the standard deviation ratio target, $\rho$ is the correlation target and $\delta$ is the bias target between the model and the observation dataset).

The behavior of the Spatial Pattern Efficiency Metric ($E_{sp}$) (Fig. 4) is very similar to that of SPAEF. There is good performance in the case of no bias, across the values of the correlation coefficient, but it performs worse as the bias increases, especially for the case of a high correlation coefficient ($\rho = 0.9$), as the minimum of the curves shifts to standard deviation values greater than 1. Specifically, for the cases of a small bias ($\delta = 1.0$) the metric performs well for the normally distributed data, but for the skewed data, the minimum of the curves shifts to standard deviations greater than 1. Conversely, for the cases of a large bias ($\delta = 3.0$), while the skewed variables have their minimum shifted to standard deviation ratios larger than 1 for all correlation coefficients, the normally distributed cases also have a shift of the minimum to standard deviations greater than one that is perceivable for the high correlation coefficient only ($\rho = 0.9$). This metric, like in the case of SPAEF, is bias-insensitive, since different bias values do not directly reflect to changes in the values of the metric.

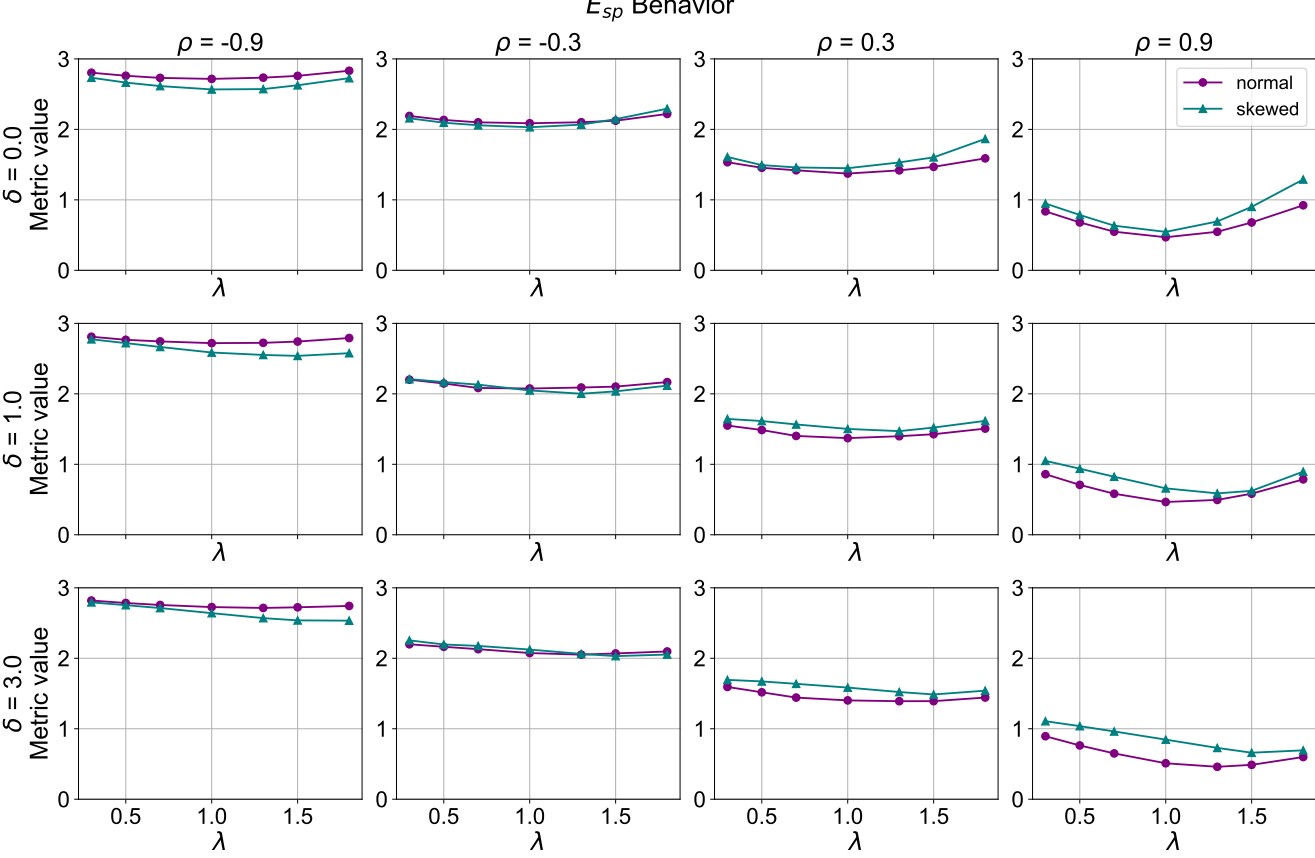

**Figure 4.** $E_{sp}$ metric behavior ($\lambda$ is the standard deviation ratio target, $\rho$ is the correlation target and $\delta$ is the bias target between the model and the observation dataset).

On the other hand, the WSPAEF metric (Fig. 5) exhibits a distinct behavior compared to the aforementioned metrics. Similarly to the two previous metrics, a good performance is observed for both the normal and skewed distributed data in the absence of bias. However, for the normally distributed cases across all combinations of correlation coefficient and bias, the minimum value of the curve is consistently at a standard deviation value of 1. For the skewed data, in the cases where there is a small bias present ($\delta = 1.0$) and the correlation coefficient is near 0 ($\rho = -0.3, 0.3$), the minimum value of the curve shifts at standard deviations ratio slightly below 1. When a substantial bias is present ($\delta = 3.0$), the minimum value of the curve for the skewed distributed data shifts to a standard deviation ratio significantly below 1. Additionally, unlike SPAEF and $E_{sp}$, this metric demonstrates sensitivity to bias, as the metric values generally increase with increasing values of bias.


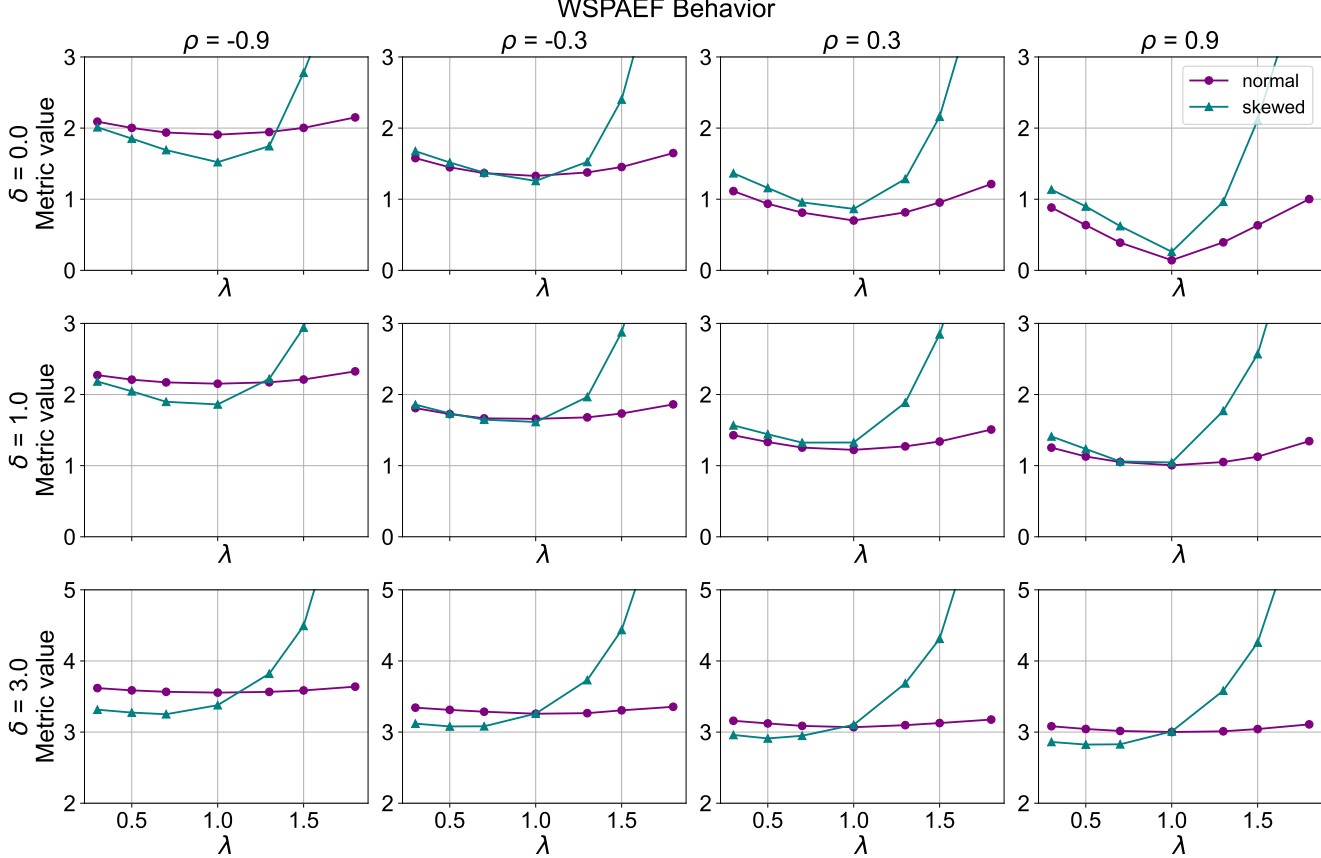

**Figure 5.** WSPAEF metric behavior ($\lambda$ is the standard deviation ratio target, $\rho$ is the correlation target and $\delta$ is the bias target between the model and the observation dataset).

The MSPAEF metric, illustrated in Fig. 6, demonstrates a robust performance across all combinations of correlation coefficients and biases for both the normal and skewed distributed variables. Nonetheless, under conditions of large bias ($\delta = 3.0$) and negative correlation coefficients ($\rho = -0.9, -0.3$), the response curve for normally distributed data exhibits a near-plateau around the standard deviation ratio of 1. This suggests a diminished sensitivity to variations in the standard deviation ratio within this specific range. Similar to WSPAEF, MSPAEF is sensitive to bias, with values generally increasing with increasing bias. This trend is particularly pronounced for cases with a large positive correlation coefficient ($\rho = 0.9$). Generally, the metric values decrease with increasing correlation and decreasing bias, which aligns with the expected behavior of a well-behaved metric.

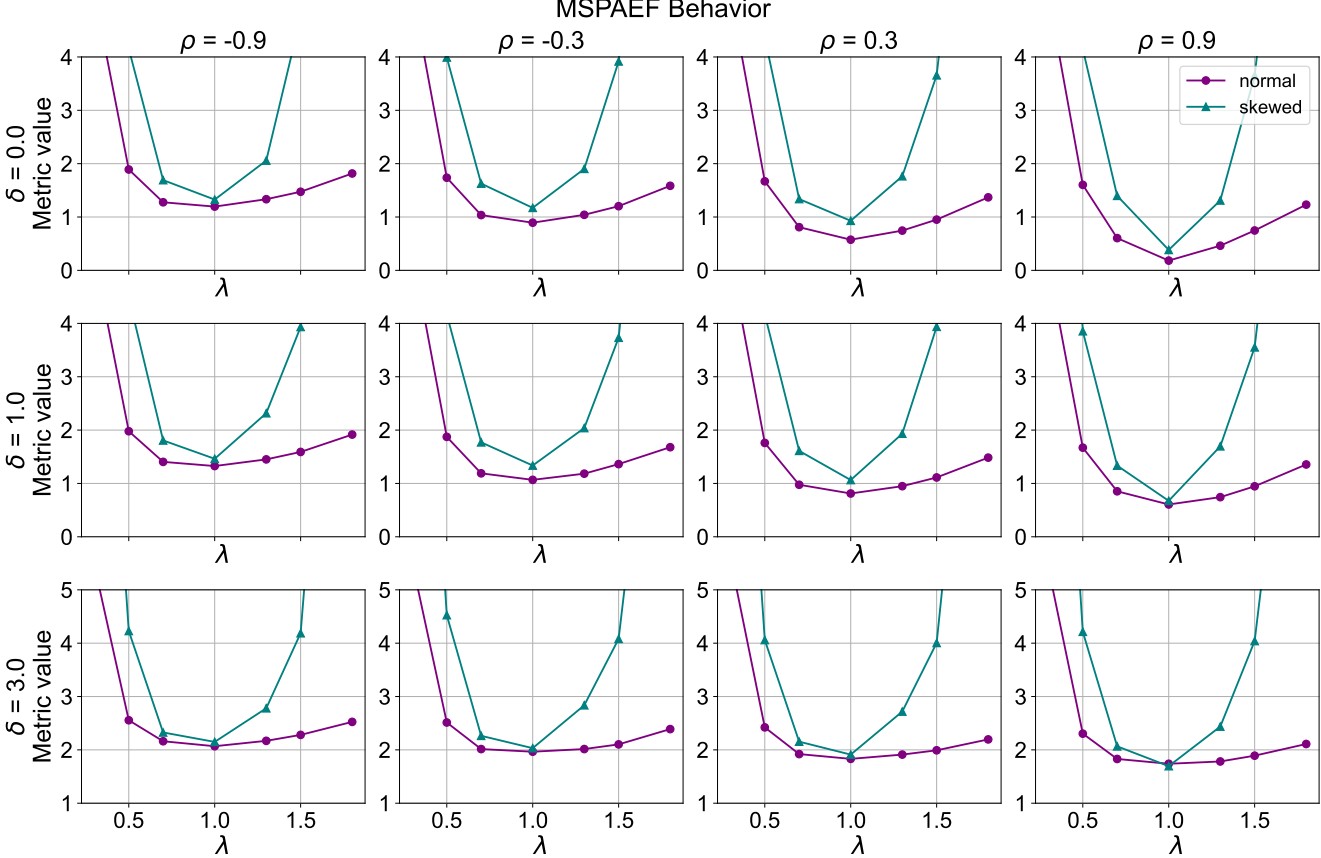

**Figure 6.** MSPAEF metric behavior ($\lambda$ is the standard deviation ratio target, $\rho$ is the correlation target and $\delta$ is the bias target between the model and the observation dataset).

## 3.2 Synthetic data examples

Two examples are presented to illustrate the differences in the values and the interpretation of the four metrics, using normally distributed synthetic data as described in the Methods section. These synthetic datasets are designed to realistically resemble
observations and climate model output for evaluation. The first example shown in Fig. 7a, presents the synthetic observations and two different models with predefined statistical properties. The observations were generated with a mean value of 200 and a standard deviation of 70, representing a scenario such as the mean annual precipitation in millimeters (mm) in a specific region of the planet. Although daily precipitation often follows a highly skewed or exponential distribution, annual averages can be closely approximated to a normal distribution, especially outside of polar regions (see Appendix B). Model A was created using
a spatial correlation coefficient equal to 0.85, a standard deviation ratio of 1.1 and a domain-average bias of -10. In contrast, Model B was created using a correlation coefficient of -0.4, a standard deviation ratio of 1.0, and a domain-average bias of

0. While the use of negative spatial correlations might look unusual, especially at global scales, they can occur regionally, especially in precipitation fields due to known large biases near the equator, such as the double ITCZ bias (Ma et al., 2023). M

Model A shows a spatial pattern that resembles the observations closely, with only minor differences. In contrast, Model B displays a significantly different spatial pattern for the variable under evaluation. Intuitively, Model A should be considered the better model due to its close alignment with the observations and its minimal relative bias. On the other hand, Model B should be regarded as the less skillful model because its spatial pattern diverges considerably from the observations, even though it does not exhibit significant bias when looking at the domain average.

In Fig. 7b, the boxplots show the distribution of the values of the four metrics from 20 different realizations of the synthetic data generated with the aforementioned parameters. The WSPAEF metric indicates that Model B has better performance compared to Model A, since it has the lower values for the metric, while the other three metrics conclude that Model A illustrates a model with better performance compared to Model B, as our intuition suggests. The bad performance of WSPAEF in this example is attributed to its very high sensitivity to the actual values of the bias due to the Wasserstein distance component, which overwhelms the contribution of the other two components on the final value of the metric.

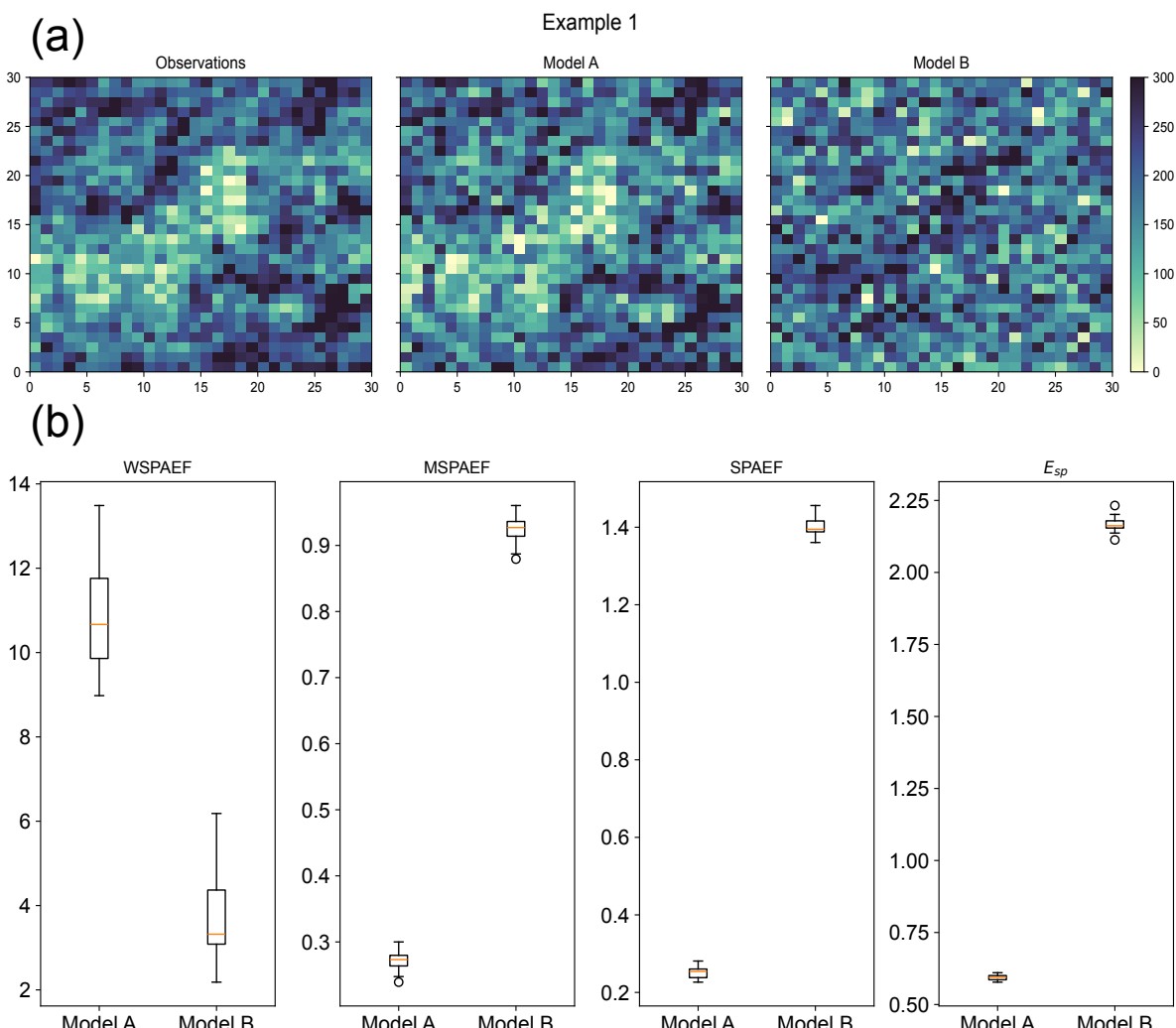

**Figure 7.** (a) Example of spatial pattern of synthetic observations and Models A and B in Example 1 (Observations were to represent a scenario of the mean annual precipitation in mm of a specific region. Model A represents a model with high correlation and a small relative bias compared to the observations, while Model B represents a model with negative correlation and no bias compared to the observations), (b) boxplots of the metric values for Models A and B. (SPAEF: Spatial Efficiency, WSPAEF: Wasserstein Spatial Efficiency, MSPAEF: Modified Spatial Efficiency, $E_{sp}$: Spatial Pattern Efficiency)

The second example is shown in Fig. 8. The observations of this example were generated with a mean of 6 and a standard deviation of 1, representing a scenario such as the annual temperature in a region of the planet. Model A was created using a spatial correlation coefficient of 0.6, a standard deviation ratio of 1.1 and a domain-average bias of 0. Model B was created using a spatial correlation coefficient of 0.8, a standard deviation ratio of 2.0 and a domain-average bias of 7.5. While this large bias example might seem unrealistic at global scales, it can occur at local scales and/or momentarily (McSweeney et al., 2015).

Model A spatial pattern exhibits some correlation compared to the original data but with no significant bias present. In contrast, Model B has a very similar spatial pattern for the variable, but with a very pronounced positive bias. Based on these observations, intuitively, Model A should be considered the better of the two models. Although its correlation with the observations is average, the lack of bias makes it more reliable. On the other hand, Model B should be considered the worst of the two because, despite its similarity to the observed spatial pattern, it has a substantial bias compared to the observations.

In Fig. 8b the boxplots show the distribution of the values of the four metrics from 20 different realizations of the synthetic data generated with the aforementioned parameters. The WSPAEF and MSPAEF metrics indicate that the Model A has far better performance compared to the Model B. This is expected, since there is a large bias present in Model B which dramatically increases the values of these two bias-sensitive metrics, contrariwise to Model A which has no bias present. On the other hand, the other two metrics indicate that Model B illustrates a model with better performance compared to Model A, because these

two metrics are bias-insensitive and therefore are affected more by the underlying patterns of the variable in the two models.

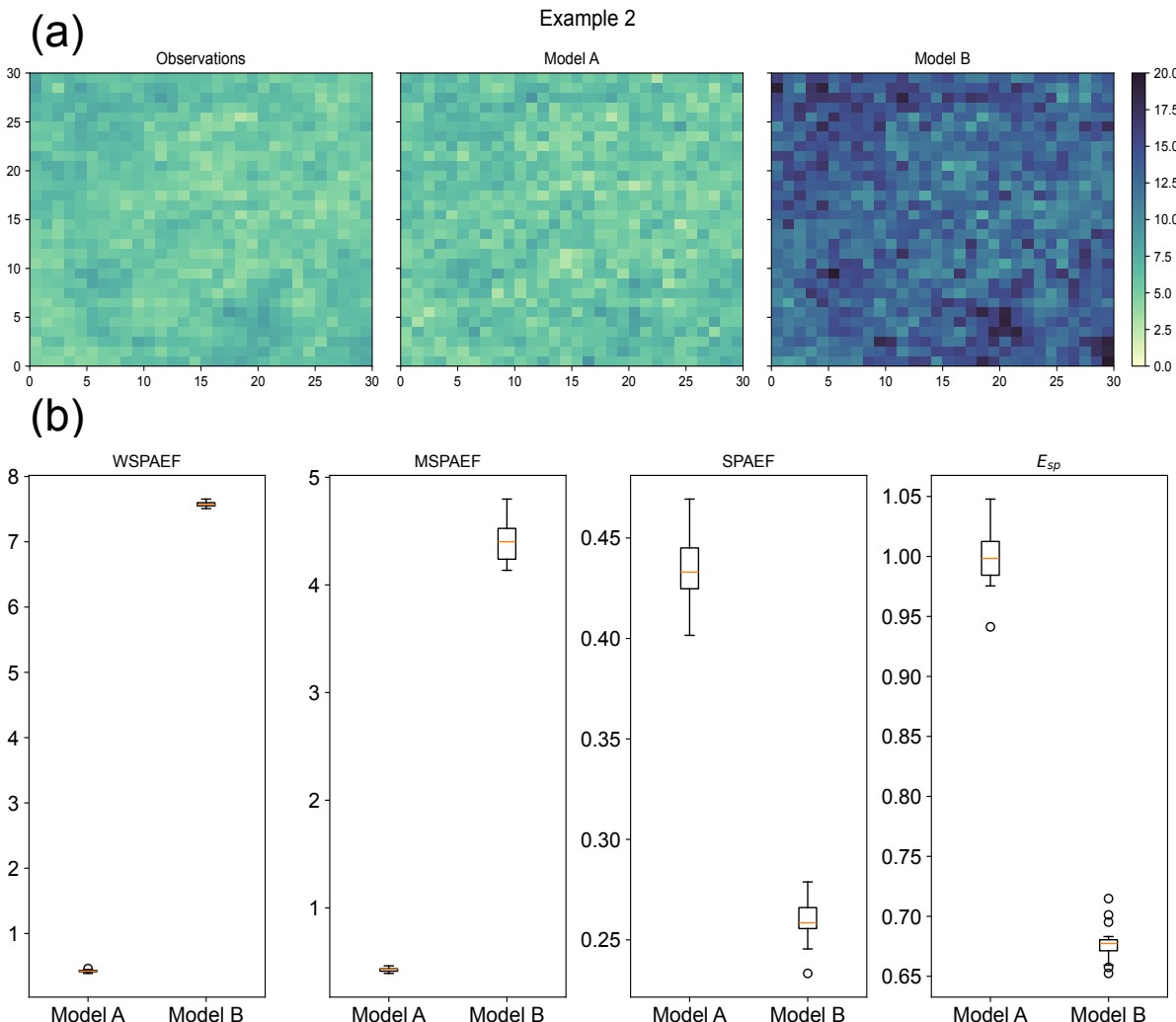

**Figure 8.** (a) Example of the spatial pattern of observations and Models A and B in Example 2 (Observations were to represent a scenario of the annual temperature in degrees Celsius of a specific region. Model A represents a model with medium correlation and no bias compared to the observations, while Model B represents a model with high correlation and a large bias compared to the observations), (b) boxplots of the metric values for Models A and B. (SPAEF: Spatial Efficiency, WSPAEF: Wasserstein Spatial Efficiency, MSPAEF: Modified Spatial Efficiency, $E_{sp}$: Spatial Pattern Efficiency)

## 3.3 Application to real data

In this section, the four metrics were used in their original form, where the value of 1 indicates the best performance for MSPAEF, SPAEF and $E_{sp}$, while for WSPAEF the best performance is indicated at a value of 0. The metrics were applied to global CMIP6 model output, in order to benchmark them against the ERA5 reanalysis for annual precipitation totals and mean near-surface temperature. For both examples, we averaged over the 1981-2010 period. These multi-decadal averages were


used to reduce the short term variability, and highlight the long term climatological signal (Du et al., 2022; Nooni et al., 2023). The performance of the CMIP6 models using the MSPAEF metric for the two variables is presented in Fig. 9. Some models perform very well for the spatial distribution of both variables (e.g., CESM2), while others perform well for one of the two variables, and much poorer for the other. For instance, the MIROC6 model is found to perform better than most other CMIP6 models for precipitation, but it is the second worst-performing model in representing the spatial distribution of near-surface temperature. On the other hand, ACCESS-ESM1-5 performs very well for the temperature, being in the top few models, but it is the worst-performing model in representing precipitation. Then, there are models that perform badly for both variables, such as MIROC-ES2L which performs below average in terms of the precipitation variable and is the worst-performing model by far in the representation of temperature.

In line with a previous study (Jun et al., 2008), we generally expect models from the same developers or model family to cluster together in performance, and we do observe this behavior for most groups. However, a few clear outliers emerge. These outliers typically correspond to models with substantially coarser atmospheric resolution (see Appendix C), which can markedly alter precipitation characteristics and therefore degrade overall performance relative to their higher-resolution counterparts. Although differences in parameterizations and internal variability may also play a role, the systematic link between resolution and performance offers the most plausible explanation for the departures from the expected clustering.

Regarding the actual values of the metric, all CMIP6 models except MIROC-ES2L, have values greater than 0.9 for the temperature variable. MRI-ESM2-0 is the model with the best performance for the temperature variable, which indicates that this model better represents not only the spatial patterns, but also the actual values of temperature at each grid-point. Contrariwise, all models except the two CESM2s and the NorESM2-MM models have values less than 0.85 for the precipitation variable. This indicates that the CMIP6 models generally can capture well the spatial distribution and magnitude of the temperature, while they struggle a lot more in the representation of the precipitation features.

Although comparing the MSPAEF values for two different variables is not straightforward, some conclusions on the nature of the variables can be deduced. The spatial distribution of temperature is influenced by factors such as orography, latitude, and proximity to the oceans, which are easier to be represented in climate models, due to their generally smoother and slow-varying nature in space. On the other hand, precipitation tends to have steeper gradients. Modeling precipitation is significantly more complex (Legates, 2014; Räisänen, 2007), as it is largely influenced by sub-grid-scale processes like convection and cloud microphysics that are not explicitly resolved in the models, and they need to be parametrized (Pieri et al., 2015), which makes it more difficult for climate models to correctly capture the underlying gradients and the actual grid-point values.

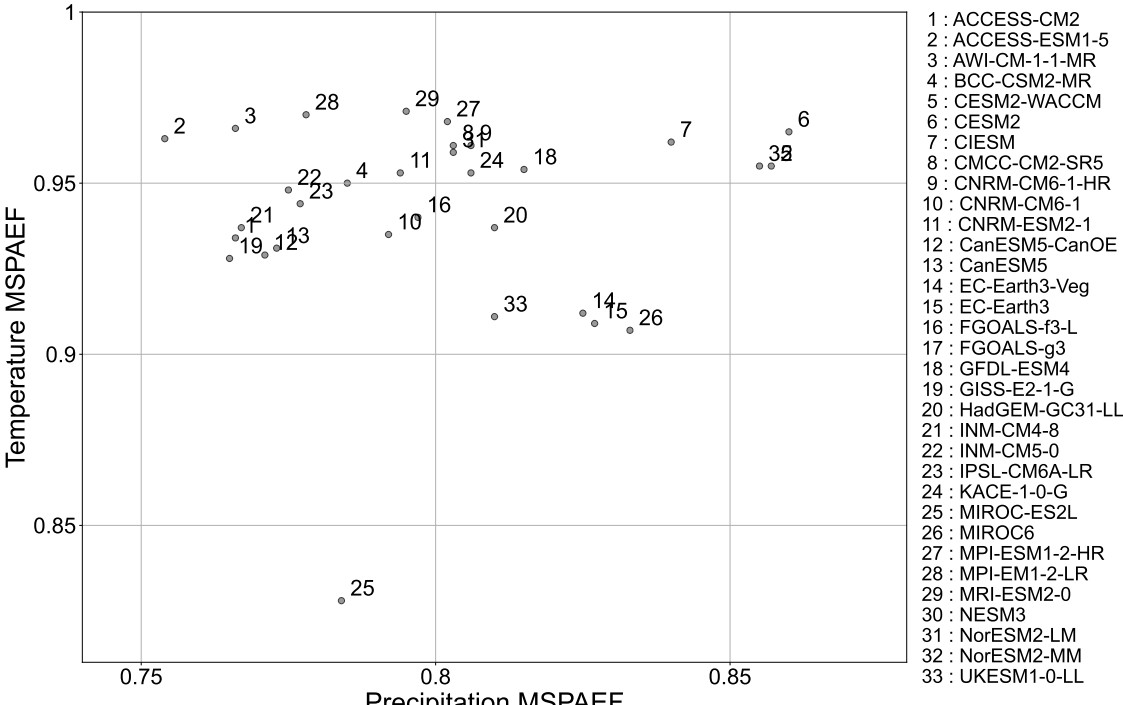

**Figure 9.** Scatter plot of the Modified Spatial Efficiency (MSPAEF) values for annual mean precipitation and 2-meter temperature derived from 33 CMIP6 models during the historical period of 1981-2010.

Tables 1 and 2 present the values and the corresponding ranking of the models, based on the four metrics discussed in this study, for annual precipitation and 2-meter temperature, respectively. The metrics generally exhibit a degree of consensus regarding the identification of the best and worst-performing models for each variable. However, the precise ranking order of the models varies across the different metrics.

For precipitation (Table 1), the model rankings derived from the WSPAEF metric differ the most from those obtained from the other three metrics, as illustrated in Fig. 10. The normalized sum of absolute rank differences between the CMIP6 models exceeds the 7 units when comparing the WSPAEF metric with the other three metrics. This significant differentiation in the WSPAEF rankings can be attributed to the very large values this metric attains as a consequence of its high sensitivity to the inherent biases that arise from its Wasserstein distance component. Given that annual precipitation was the variable under consideration, a bias of just 30 mm in the globally averaged total annual precipitation can result in WSPAEF metric values exceeding the 30 units, even if this bias represents only approximately 3% of the total annual global precipitation.

The MSPAEF metric rankings exhibit the most similarity to those of $E_{sp}$ with a rank difference value of 2.8 and the least similarity to WSPAEF with a rank difference value of 7.9, with SPAEF demonstrating an intermediary level of similarity.

This suggests that MSPAEF effectively captures the spatial distribution of precipitation while at the same time not being overwhelmed by existing bias.

For temperature (Table 2), the four metrics generally exhibit a greater degree of agreement regarding the ranking of the models. However, more variability is observed in the identification of the top-performing models. Concerning the WSPAEF metric in Table 2, its values exceed the 2 units only for three models. This is attributed to the considerably smaller absolute values of the mean bias between the individual models and the observations, measured in Kelvin, which rarely exceed one or two degrees. This relatively small absolute value of the mean bias allows the other components of the WSPAEF metric, like the correlation coefficient, to exert a greater influence in the final metric value. This enables WSPAEF to achieve greater consistency in its rankings with the other three metrics for temperature than for precipitation, as depicted in Fig. 10.

The most substantial differences in temperature rankings are observed between WSPAEF and SPAEF, while the smallest difference occurs between the $E_{sp}$ and SPAEF metric. The rankings of MSPAEF exhibit the greatest similarity to the $E_{sp}$ metric with rank differences value of 3.8, and the least similarity to SPAEF with rank differences value of 5.5.

The similarity of MSPAEF to the bias-insensitive metrics might look unusual at first glance (compared to the other bias-sensitive metrics), given that MSPAEF is a bias-sensitive metric. However, this is less surprising when considering the large sensitivity of WSPAEF (the other bias-sensitive metric) to the absolute value of the mean bias. This sensitivity is what causes WSPAEF's performance to diverge significantly from that of MSPAEF (and the other two bias-insensitive metrics), particularly for the precipitation variable, where the bias in the mean can be large in absolute terms.

Using weights for the different components could serve as a way to improve the performance of the existing metrics, to closely match the performance of MSPAEF. For example, in the case of WSPAEF, achieving a behavior more consistent with MSPAEF in the presence of significant absolute mean bias would require reducing the relative contribution of the WD component. By contrast, SPAEF and $E_{sp}$ already perform similarly to MSPAEF when bias values are small, but as bias increases, their lack of bias-sensitive components will limit their ability to achieve similar performance to MSPAEF.

While in this demonstration we evaluate and rank models separately for each variable and each metric, in many real-world applications, the overall model performance can be assessed using multiple variables. There are many multi-criterion model ranking techniques that can do this, such as Compromise Programming (CP) (Refaey et al., 2019; Baghel et al., 2022), Technique for Order Preference by Similarity to an Ideal Solution (TOPSIS) (Raju and Kumar, 2015) and Cooperative Game Theory (Gershon and Duckstein, 1983).

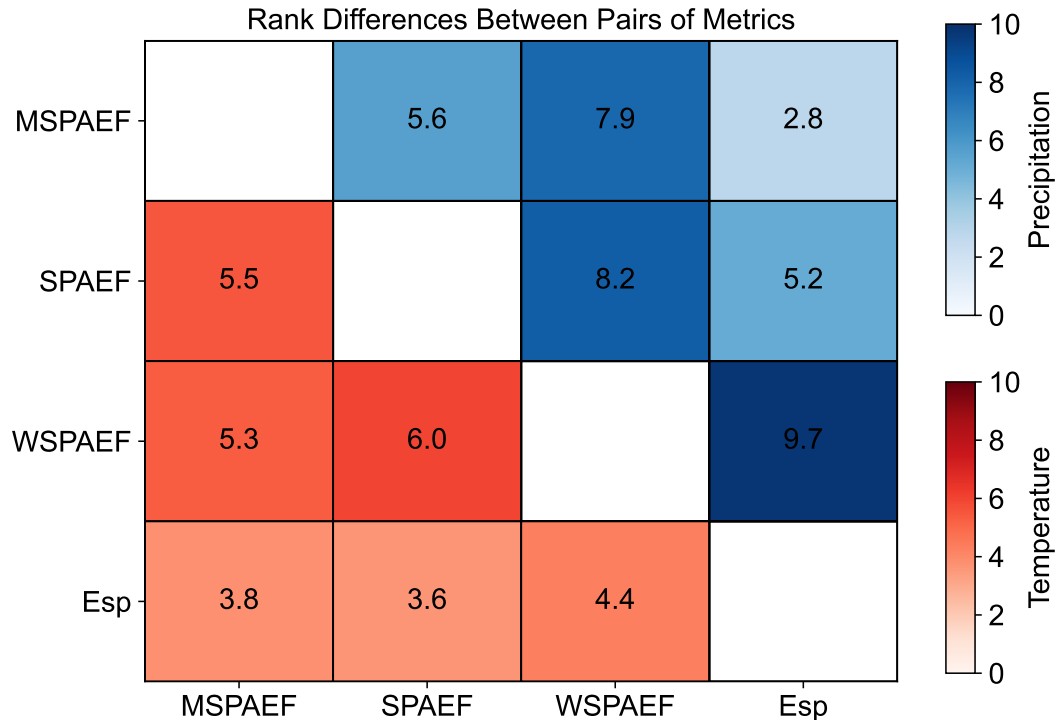

**Figure 10.** Normalized sum of absolute rank differences between pairs of metrics for the 33 CMIP6 models, based on mean annual precipitation and temperature during the historical period of 1981–2010. (SPAEF: Spatial Efficiency, WSPAEF: Wasserstein Spatial Efficiency, MSPAEF: Modified Spatial Efficiency, $E_{sp}$: Spatial Pattern Efficiency)

## 4   Conclusions

435   Model evaluation plays a vital role in climate model development and is crucial for understanding the limitations of these models, especially before conducting impact assessments. In this study, we present an inter-comparison exercise of several performance metrics that focus on the spatial representation of model output.

The bias-insensitive metrics, SPAEF and $E_{sp}$ perform well in the cases of relatively low biases. With increasing bias, their performance generally deteriorates, something that is evident by the shift of the curve's minimum to values of standard
440   deviation ratios greater than 1. Contrariwise, the WSPAEF metric generally has a better all-round performance than the two bias-insensitive metrics. Nonetheless, it performs poorly for skewed distributed variables as the bias increases.

To address these limitations of the existing metrics, we introduced a new metric, the Modified Spatial Efficiency (MSPAEF). MSPAEF has shown robust performance across a range of correlation, bias, and standard deviation ratio values, for variables that follow either a normal or a skewed distribution. Nevertheless, it has a slightly reduced sensitivity to changes in the standard
445   deviation ratio near 1 when a large bias is present, particularly for normally distributed variables. This reduced sensitivity arises

**Table 1.** Values of the four metrics and CMIP6 model rank for the annual total precipitation, averaged over the 1981-2010 period. (SPAEF: Spatial Efficiency, WSPAEF: Wasserstein Spatial Efficiency, MSPAEF: Modified Spatial Efficiency, $E_{sp}$: Spatial Pattern Efficiency)

| Model ID | MSPAEF | SPAEF | WSPAEF | $E_{sp}$ | Rank MSPAEF | Rank SPAEF | Rank WSPAEF | Rank $E_{sp}$ |
|---|---|---|---|---|---|---|---|---|
| 1 | 0.766 | 0.866 | 64.057 | 0.554 | 28 | 15 | 29 | 15 |
| 2 | 0.754 | 0.901 | 89.795 | 0.589 | 31 | 5 | 31 | 10 |
| 3 | 0.766 | 0.816 | 34.66 | 0.494 | 28 | 27 | 18 | 26 |
| 4 | 0.785 | 0.86 | 18.115 | 0.5 | 20 | 16 | 4 | 25 |
| 5 | 0.857 | 0.919 | 15.979 | 0.658 | 2 | 1 | 2 | 2 |
| 6 | 0.86 | 0.918 | 13.463 | 0.665 | 1 | 2 | 1 | 1 |
| 7 | 0.84 | 0.911 | 34.8 | 0.634 | 4 | 4 | 19 | 4 |
| 8 | 0.803 | 0.9 | 49.172 | 0.578 | 13 | 6 | 25 | 11 |
| 9 | 0.806 | 0.835 | 27.046 | 0.571 | 11 | 22 | 10 | 13 |
| 10 | 0.792 | 0.853 | 25.821 | 0.545 | 19 | 17 | 8 | 17 |
| 11 | 0.794 | 0.848 | 28.921 | 0.534 | 18 | 18 | 12 | 20 |
| 12 | 0.771 | 0.802 | 53.156 | 0.494 | 26 | 29 | 27 | 26 |
| 13 | 0.773 | 0.802 | 52.572 | 0.503 | 25 | 29 | 26 | 24 |
| 14 | 0.825 | 0.833 | 32.86 | 0.625 | 7 | 23 | 15 | 6 |
| 15 | 0.827 | 0.833 | 33.682 | 0.625 | 6 | 23 | 17 | 6 |
| 16 | 0.797 | 0.874 | 25.285 | 0.525 | 16 | 11 | 6 | 21 |
| 17 | 0.724 | 0.715 | 96.786 | 0.423 | 32 | 33 | 33 | 33 |
| 18 | 0.815 | 0.842 | 31.245 | 0.578 | 8 | 20 | 13 | 11 |
| 19 | 0.765 | 0.805 | 35.667 | 0.453 | 30 | 28 | 20 | 31 |
| 20 | 0.81 | 0.898 | 47.345 | 0.61 | 9 | 7 | 23 | 8 |
| 21 | 0.767 | 0.818 | 31.772 | 0.463 | 27 | 26 | 14 | 30 |
| 22 | 0.775 | 0.826 | 27.702 | 0.477 | 24 | 25 | 11 | 28 |
| 23 | 0.777 | 0.87 | 37.813 | 0.521 | 23 | 13 | 22 | 22 |
| 24 | 0.806 | 0.873 | 37.074 | 0.56 | 11 | 12 | 21 | 14 |
| 25 | 0.784 | 0.783 | 90.885 | 0.508 | 21 | 31 | 32 | 23 |
| 26 | 0.833 | 0.842 | 79.377 | 0.632 | 5 | 20 | 30 | 5 |
| 27 | 0.802 | 0.87 | 26.054 | 0.538 | 15 | 13 | 9 | 19 |
| 28 | 0.778 | 0.846 | 32.896 | 0.477 | 22 | 19 | 16 | 28 |
| 29 | 0.795 | 0.878 | 25.41 | 0.548 | 17 | 9 | 7 | 16 |
| 30 | 0.716 | 0.753 | 55.362 | 0.429 | 33 | 32 | 28 | 32 |
| 31 | 0.803 | 0.878 | 17.043 | 0.543 | 13 | 9 | 3 | 18 |
| 32 | 0.855 | 0.917 | 22.585 | 0.655 | 3 | 3 | 5 | 3 |
| 33 | 0.81 | 0.885 | 48.338 | 0.61 | 9 | 8 | 24 | 8 |

from the dominant influence of the large bias, primarily due to the NRMSE and relative bias components, in the final metric value.

The synthetic data examples of Sect. 3.2, demonstrate that bias-insensitive metrics are suboptimal for model evaluation when bias is a critical factor, with this limitation being particularly evident in Example 2. (Fig. 8). Conversely, for variables with 450 large absolute values, even a small bias relative to the mean observational value can have a substantial impact on the WSPAEF metric. In contrast, the MSPAEF and the two bias-insensitive metrics are less affected, as demonstrated in Example 1 (Fig. 7). Thus, in the two examples presented in Sect. 3.2, MSPAEF was the only metric whose values consistently aligned with the

**Table 2.** Values of the four metrics and CMIP6 model rank for the monthly 2m temperature variable averaged over the 1981-2010 period. (SPAEF: Spatial Efficiency, WSPAEF: Wasserstein Spatial Efficiency, MSPAEF: Modified Spatial Efficiency, $E_{sp}$: Spatial Pattern Efficiency)

| Model ID | MSPAEF | SPAEF | WSPAEF | $E_{sp}$ | Rank MSPAEF | Rank SPAEF | Rank WSPAEF | Rank $E_{sp}$ |
|---|---|---|---|---|---|---|---|---|
| 1 | 0.934 | 0.861 | 0.772 | 0.888 | 24 | 23 | 13 | 22 |
| 2 | 0.963 | 0.857 | 1.06 | 0.896 | 6 | 26 | 20 | 18 |
| 3 | 0.966 | 0.921 | 0.404 | 0.93 | 4 | 4 | 1 | 1 |
| 4 | 0.95 | 0.86 | 1.172 | 0.884 | 16 | 24 | 25 | 25 |
| 5 | 0.955 | 0.935 | 0.539 | 0.925 | 11 | 2 | 4 | 4 |
| 6 | 0.965 | 0.936 | 0.543 | 0.926 | 5 | 1 | 5 | 3 |
| 7 | 0.962 | 0.908 | 1.226 | 0.918 | 7 | 7 | 29 | 7 |
| 8 | 0.961 | 0.892 | 1.074 | 0.902 | 8 | 14 | 21 | 15 |
| 9 | 0.961 | 0.904 | 1.113 | 0.905 | 8 | 9 | 23 | 12 |
| 10 | 0.935 | 0.876 | 1.117 | 0.904 | 23 | 16 | 24 | 13 |
| 11 | 0.953 | 0.876 | 0.744 | 0.903 | 14 | 16 | 11 | 14 |
| 12 | 0.929 | 0.846 | 1.088 | 0.878 | 26 | 28 | 22 | 27 |
| 13 | 0.931 | 0.85 | 1.006 | 0.879 | 25 | 27 | 18 | 26 |
| 14 | 0.912 | 0.871 | 1.181 | 0.867 | 28 | 20 | 27 | 29 |
| 15 | 0.909 | 0.872 | 1.224 | 0.868 | 30 | 19 | 28 | 28 |
| 16 | 0.94 | 0.888 | 0.969 | 0.89 | 20 | 15 | 17 | 21 |
| 17 | 0.855 | 0.818 | 2.001 | 0.806 | 32 | 32 | 31 | 32 |
| 18 | 0.954 | 0.893 | 0.641 | 0.921 | 13 | 13 | 8 | 6 |
| 19 | 0.928 | 0.896 | 0.886 | 0.894 | 27 | 12 | 16 | 19 |
| 20 | 0.937 | 0.859 | 0.864 | 0.899 | 21 | 25 | 15 | 17 |
| 21 | 0.937 | 0.841 | 1.174 | 0.86 | 21 | 31 | 26 | 30 |
| 22 | 0.948 | 0.87 | 1.025 | 0.888 | 17 | 22 | 19 | 22 |
| 23 | 0.944 | 0.912 | 0.776 | 0.912 | 18 | 6 | 14 | 8 |
| 24 | 0.953 | 0.874 | 0.653 | 0.9 | 14 | 18 | 9 | 16 |
| 25 | 0.828 | 0.793 | 2.187 | 0.788 | 33 | 33 | 33 | 33 |
| 26 | 0.907 | 0.844 | 2.034 | 0.847 | 31 | 29 | 32 | 31 |
| 27 | 0.968 | 0.914 | 0.529 | 0.929 | 3 | 5 | 3 | 2 |
| 28 | 0.97 | 0.897 | 0.546 | 0.91 | 2 | 11 | 6 | 10 |
| 29 | 0.971 | 0.905 | 0.462 | 0.925 | 1 | 8 | 2 | 4 |
| 30 | 0.943 | 0.9 | 0.594 | 0.891 | 19 | 10 | 7 | 20 |
| 31 | 0.959 | 0.926 | 0.751 | 0.911 | 10 | 3 | 12 | 9 |
| 32 | 0.955 | 0.871 | 0.693 | 0.908 | 11 | 20 | 10 | 11 |
| 33 | 0.911 | 0.843 | 1.341 | 0.887 | 29 | 30 | 30 | 24 |

intuitive assessment of model performance for both cases. This is attributed to MSPAEF being a bias-sensitive metric, unlike SPAEF and $E_{sp}$, while also accounting for the relative rather than absolute value of the mean bias, unlike the WSPAEF metric.

455 Additionally, a distinguishing characteristic of MSPAEF is its scale-independence, meaning it is unaffected by the units of the variable, unlike the other three metrics.

In the real-world application of Sect. 3.3 using global CMIP6 models, it becomes obvious that for variables that can have large absolute value of bias, such as the total annual precipitation, the WSPAEF metric is greatly affected by it and it differs the most in the ranking compared to the other three metrics. The differences of WSPAEF with the other metrics are substantially

reduced with the temperature variable, due to the smaller absolute values of the mean bias, which allows its the other components to have a larger contribution to the final value of the metric. The MSPAEF rankings are the most similar to those of $E_{sp}$ for both variables, highlighting the ability of MSPAEF to evaluate spatial patterns. However, some differences in the rankings arise due to MSPAEF being a bias-sensitive metric.

Through the use of appropriate weights, the existing metrics can be adjusted to better align with the performance of MSPAEF. Nevertheless, on many occasions, such as when there is insignificant bias, their original unweighted forms often perform sufficiently well, reducing the need for such modifications.

The MSPAEF metric is not proposed as a substitute for multi-metric evaluation, which remains essential in climate model assessment. It was designed as a comprehensive similarity measure that incorporates both spatial pattern agreement and bias. It is defined in a way that emphasizes spatial pattern similarity when the relative bias is small, but it increasingly emphasizes the bias as it becomes more significant. This design allows MSPAEF to act as a balanced indicator, adapting its emphasis depending on the characteristics of the data. While this metric was developed for evaluating gridded climate model output, its underlying rationale and its inherent flexibility make it suitable for assessing the performance of other types of geoscientific or environmental models where the spatial distribution of simulated variables is expected to follow certain patterns in space. Its use alongside established measures can help provide a clearer and more consistent picture of model performance.

## Appendix A:  Modification of Metrics

The SPAEF, $E_{sp}$, and MSPAEF metrics were modified to take values equal to or greater than zero, for the inter-comparison of the metrics.

Specifically, the SPAEF metric was modified as follows:

$$SPAEF = \sqrt{(\alpha - 1)^2 + (\beta - 1)^2 + (\gamma - 1)^2} \tag{A1}$$

The $E_{sp}$ metric was modified as follows:

$$E_{sp} = \sqrt{(r_s - 1)^2 + (\gamma - 1)^2 + (\alpha - 1)^2} \tag{A2}$$

The MSPAF metric was modified as follows:

$$MSPAEF = \frac{1}{\sqrt{4}} \sqrt{(\alpha - 1)^2 + (\beta)^2 + (\gamma)^2 + (\delta)^2} \tag{A3}$$

## Appendix B: Probability Distributions of Precipitation from EC-Earth3 CMIP6 Historical Run

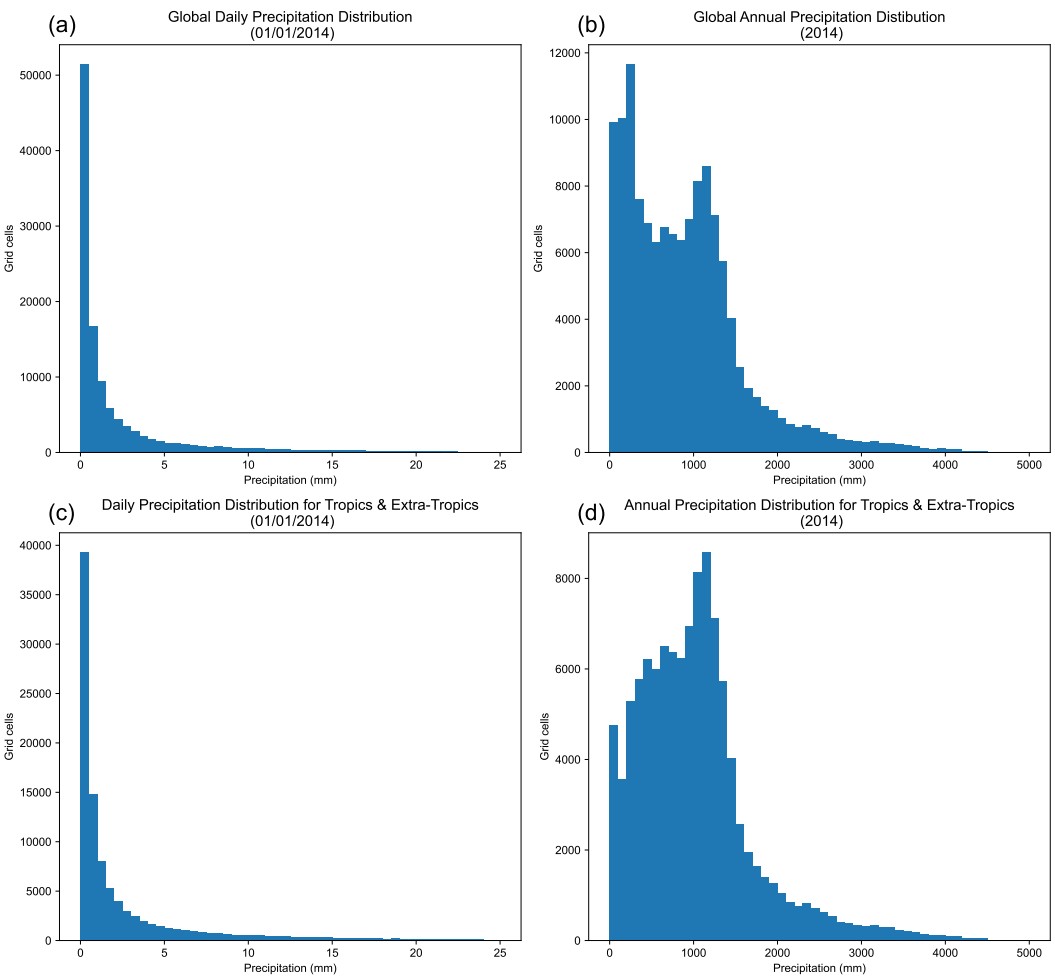

**Figure B1.** Probability distributions from the r1i1p1f1 variant of the historical run of the EC-Earth3 CMIP6 model. (a) Global daily precipitation for 1 January 2014. (b) Global annual precipitation for 2014. (c) Daily precipitation for the Tropics and Extra-tropics on 1 January 2014. (d) Annual precipitation for the Tropics and Extra-tropics in 2014.

 **Appendix C: CMIP6 models**

**Table C1.** CMIP6 models

| Model ID | Model name | Parent Organization | Atmospheric resolution | Ocean Resolu- tion | References |
|---|---|---|---|---|---|
| 1 | ACCESS-CM2 | CSIRO-ARCCSS | 250km | 100km | Dix et al. (2023) |
| 2 | ACCESS-ESM1-5 | CSIRO | 250km | 100km | Ziehn et al. (2023) |
| 3 | AWI-CM-1-1-MR | AWI | 100km | 25km | Semmler et al. (2023) |
| 4 | BCC-CSM2-MR | BCC | 100km | 50km | Xin et al. (2023) |
| 5 | CESM2-WACCM | NCAR | 100km | 100km | Danabasoglu (2023b) |
| 6 | CESM2 | NCAR | 100km | 100km | Danabasoglu (2023a) |
| 7 | CIESM | THU | 100km | 50km | Huang (2023) |
| 8 | CMCC-CM2-SR5 | CMCC | 100km | 100km | Lovato and Peano (2023) |
| 9 | CNRM-CM6-1-HR | CNRM-CERFACS | 100km | 25km | Voldoire (2023b) |
| 10 | CNRM-CM6-1 | CNRM-CERFACS | 250km | 100km | Voldoire (2023a) |
| 11 | CNRM-ESM2-1 | CNRM-CERFACS | 250km | 100km | Seferian (2023) |
| 12 | CanESM5-CanOE | CCCma | 500km | 100km | Swart et al. (2023b) |
| 13 | CanESM5 | CCCma | 500km | 100km | Swart et al. (2023a) |
| 14 | EC-Earth3-Veg | EC-Earth-Consortium | 100km | 100km | (EC-Earth) |
| 15 | EC-Earth3 | EC-Earth-Consortium | 100km | 100km | (EC-Earth) |
| 16 | FGOALS-f3-L | CAS | 100km | 100km | YU (2023) |
| 17 | FGOALS-g3 | CAS | 250km | 100km | Li (2023) |
| 18 | GFDL-ESM4 | NOAA-GFDL | 100km | 50km | Krasting et al. (2023) |
| 19 | GISS-E2-1-G | NASA-GISS | 250km | 100km | for Space Studies (NASA/GISS) |
| 20 | HadGEM-GC31-LL | MOHC | 250km | 100km | Ridley et al. (2023) |
| 21 | INM-CM4-8 | INM | 100km | 100km | Volodin et al. (2023a) |
| 22 | INM-CM5-0 | INM | 100km | 50km | Volodin et al. (2023b) |
| 23 | IPSL-CM6A-LR | IPSL | 250km | 100km | Boucher et al. (2023) |
| 24 | KACE-1-0-G | NIMS-KMA | 250km | 100km | Byun et al. (2023) |
| 25 | MIROC-ES2L | MIROC | 500km | 100km | Hajima et al. (2023) |
| 26 | MIROC6 | MIROC | 250km | 100km | Tatebe and Watanabe (2023) |
| 27 | MPI-ESM1-2-HR | MPI-M | 100km | 50km | Jungclaus et al. (2023) |
| 28 | MPI-ESM1-2-LR | MPI-M | 250km | 250km | Wieners et al. (2023) |
| 29 | MRI-ESM2-0 | MRI | 100km | 100km | Yukimoto et al. (2023) |
| 30 | NESM3 | NUIST | 250km | 100km | Cao and Wang (2023) |
| 31 | NorESM2-LM | NCC | 250km | 100km | Seland et al. (2023) |
| 32 | NorESM2-MM | NCC | 100km | 100km | Bentsen et al. (2023) |
| 33 | UKESM1-0-LL | MOHC | 250km | 100km | Tang et al. (2023) |

*Code and data availability.* The Python code used in this work can be obtained from https://doi.org/10.5281/zenodo.15094921 (Karpasitis, 2025). The CMIP6 data (ensemble mean of each model) can be downloaded from KNMI's Climate Explorer at https://climexp.knmi.nl/ selectfield_cmip6_knmi23.cgi? (KNMI, 2022). The ERA5 data are publicly available from the Copernicus Climate Data Store (CDS) at https://cds.climate.copernicus.eu/datasets/reanalysis-era5-single-levels-monthly-means (ECMWF, 2019)

490   *Author contributions.*   AK and GZ designed the methodology. AK performed calculations and led manuscript writing; All authors discussed the results and reviewed the paper.

*Competing interests.*   The authors declare that they have no conflict of interest

*Acknowledgements.*   This study was supported by the OptimESM project that has received funding from the European Union's Horizon Europe research and innovation programme under grant agreement No101081193.

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
