# Peer review of "A new efficiency metric for the spatial evaluation and inter-comparison of climate and geoscientific model output"

_EGUsphere, 2025_

## Author Response (AR1)

**Author's response to the Referees**

The following scheme is used to clarify our answers to the referees' comments: The reviewers' comments are denoted in black, while the author's response is denoted in blue. Any changes to the manuscript, in response to the referees' comments, are indicated as follows:

**add>>** New addition to the manuscript
**old>>** Old sentence/paragraph to be replaced (always refers to the one from the original manuscript, even if it was already changed from a previous comment)
**new>>** New sentence to replace the old one
* * *
**Community Comments (CC1)**
I enjoyed reading the manuscript. I have only two points:
1) The sentence above SPAEF equation in Koch et al, (2018) GMD paper states that "Following the multiple-component idea of KGE we present a novel spatial performance metric denoted SPAtial EFficiency (SPAEF), which was originally proposed by Demirel et al. (2018a, b)." This statement can be helpful to find the origin of SPAEF.
2) Benchmarking SPAEF with other variations of SPAEF has also been done in a 2024 SERRA paper . Table 2 lists all improved versions of SPAEF.

https://link.springer.com/content/pdf/10.1007/s00477-024-02790-4.pdf

Thank you for your feedback! We truly appreciate it.

In response to your first point, we will ensure that proper references are included for the introduction of the SPAEF metric. Additionally, we acknowledge your second point about benchmarking SPAEF against existing variations, and we will address this in the introduction.

Changes at Line 48:

old>> "*The SPAtial EFficiency metric (SPAEF) was inspired by KGE (Koch et al., 2018) and has emerged as a promising tool for evaluating hydrological models by considering three key aspects of spatial pattern accuracy: correlation coefficient, relative variability, and distribution of values in standardized space.*"

new>> "*The SPAtial EFficiency metric (SPAEF) proposed by Demirel et al. (2018b, a) was inspired by KGE (Koch et al., 2018) and has emerged as a promising tool for evaluating hydrological models by considering three key aspects of spatial pattern accuracy: correlation coefficient, relative variability, and distribution of values in standardized space.*"

Addition at the end of Line 68:

add>> "*The SPAEF metric and some of its proposed modifications have been comprehensively evaluated by Yorulmaz et al. (2024)*"

Changes at Line 84:

old>> "*The SPAEF (SPAtial EFficiency) metric by Koch et al. (2018) is a robust metric that was originally created for hydrological model evaluation, and it is used to characterize the performance of a model regarding the spatial distribution of a variable.*"

new>> "*The SPAEF (SPAtial EFficiency) metric by Demirel et al. (2018b) is a robust metric that was originally created for hydrological model evaluation, and it is used to characterize the performance of a model regarding the spatial distribution of a variable.*"
* * *
**Anonymous Referee #1 (RC1)**

This is a well-written and well-structured paper in which the authors propose a new indicator for comparing climate and geoscientific models outputs. The inclusion of the code is appreciated and contributes to open science by facilitating reproducibility. However, I believe that some revisions or additional explanations are necessary, particularly regarding the illustration of performance using synthetic data and the real-world ranking exercise. My comments are detailed below:

We would like to express our sincere gratitude to you for the time and effort invested in providing constructive feedback. We have incorporated all recommendations that we believe have significantly enhanced our manuscript. A detailed, point-by-point response to the referee's comments follows.

**Major Comment 1**

Please review the use of the term "metric" throughout the manuscript. From a mathematical perspective, not every indicator discussed may satisfy the formal properties required to be considered a metric. To prevent misunderstandings, please use other terms or consider including a brief footnote or definition clarifying your use of the term "metric".

Thank you for your important observation. We have added the following clarification in the introduction.

add>> "*In this paper, the term metric is used in a broad sense to refer to all indicators, statistics or distance measures that take as input two datasets and output a value that quantifies their relative performance or similarity. This includes, but is not limited to, quantities that satisfy the formal definition of a metric.*"

**Major Comment 2**

Line 42 The statement is too general. It would be helpful to specify in which contexts or under which conditions these "metrics" perform well, and in which they do not. Phrases such as "inhomogeneous spatiotemporal distribution" and "different statistical properties" are vague without further elaboration or examples.

We agree with the reviewer. We have revised the text and now include more specific examples to better demonstrate the contexts in which some commonly used simple metrics perform well or poorly. The revised statement reads as follows:

old>> "*While these metrics may perform well for certain applications, their effectiveness can vary significantly when applied to parameters with inhomogeneous spatiotemporal distribution or differing statistical properties. Such inconsistencies can introduce challenges in conducting a comprehensive evaluation of different climate model outputs.*"

new>> "*While these metrics can be effective for certain applications, their performance can vary significantly based on the statistical and spatial characteristics of the data. For instance, the Pearson correlation coefficient is useful for capturing the linear relationship between two datasets; however, it does not account for systematic biases in the mean or differences in variance (i.e., scale differences). Similarly, other metrics like RMSE and MAE are sensitive to both magnitude and distribution of errors, but they may be disproportionately affected by a large bias in the mean, which can lead to a misrepresentation of spatial patterns. In contrast, the Kolmogorov-Smirnov test compares the underlying distributions of two datasets but lacks spatial context, which is often crucial in geoscientific modeling. These distinct limitations underscore the challenges of using traditional metrics on variables with uneven spatiotemporal distributions, like precipitation, which is often sparsely distributed in space. They also complicate comparisons between models that possess differing statistical properties, such as varying means or levels of variation.*"

**Major Comment 3**
Line 90 The explanation of the γ component is insufficient. In the original reference, K and L are defined as histograms with n bins. Here, K and L are described as probability distributions, but n is never defined. This could lead readers to misinterpret the formulation as involving n distributions K and L, which are then summed. Please define n and ensure consistency with the source.

Thank you for this comment. We are now consistent with the source:
old>> "…, *K and L are the probability distributions of the standardized values (z-score) of the model and observations respectively,* ..."

new>> "…, *K and L are histograms with n common bins, of the standardized values (z-score) of the model and observations respectively,* …"

**Major Comment 4**
Line 112 Please clarify the definition of Φ. What does the subscript I represent, and what are the bounds of the summation? Not all readers will be familiar with the Wasserstein distance, so it's important to contextualize and adapt the mathematical notation accordingly.

Thank you for pointing out the insufficient explanation of Φ and the incomplete equation (6). We have changed $K_i$ and $L_i$ to $X_{(i)}$ and $Y_{(i)}$ in eq. (6) and modified **Line 112** to align better with the standard definition of WD:

old>> "… **WD is the Wasserstein Distance, of order p = 2. WD was calculated using the original distributions of the two datasets, to explicitly account for the bias.**"

new>> "**…, WD is the Wasserstein distance of order p=2, with X(i) and Y(i) being the i-th order statistic of the samples of the observations and model, respectively, and n is the total number of samples in each dataset. This means that the values of the two datasets have been arranged in ascending order, and X(i) is the i-th value in the sorted list. WD was calculated using the original values of the two datasets to explicitly account for the bias.**"

**Major Comment 5**

Line 141 This paragraph is somewhat redundant, as the subsequent paragraph explains the measure more clearly. If you decide to keep it, consider refining the exposition. For instance, it's stated that $\gamma$ accounts for relative bias, but then you stated that this characteristic is also influenced by $\beta$, which may confuse readers. Please clarify the distinct roles and interactions of these components.

We agree that this is redundant. We have now removed this paragraph and adjusted the text for better readability.

**Major Comment 6**

Line 200 Reintroducing all previously defined terms is redundant. You might instead state that the definitions from Equation (14) still apply, and define only the new terms. Also, would it be possible to visualize the behavior of normally distributed vs. skewed data? Including a reason for considering the skewed data scenario would also strengthen the section.

We agree with the reviewer, and we have modified the text to avoid redundancies and reintroduce some terms:

old>> "**where x is the original matrix, x_{rm} is the the original matrix after removing the mean, $\lambda$_{t} is the standard deviation ratio target, $\rho$_{t} is the correlation coefficient target, $\delta$_{t} is the bias target, $\sigma$_{x} is the standard deviation of the original matrix x and A is a matrix of the same dimensions as x, filled with random numbers drawn from the normal distribution.**"

new>> "**where x_{rm) is the original matrix after subtracting the mean, and the other terms are the same as defined in Equation (14).**"

Regarding the visualization of datasets that follow normal and skewed distributions, we have created a plot using synthetic data (see Figure 1), which shows on the left the spatial distribution of variables that follow each distribution, and on the right the corresponding histogram. The top row is for the normally distributed data, and the bottom row is for the skewed distributed data.

[Figure]

*Figure 1: Examples of synthetic data generation. Panels (a) and (b) show the spatial distribution and probability density function (PDF) respectively, of synthetic data generated with a normal distribution. Panels (c) and (d) show the corresponding spatial distribution and PDF for skewed data*

The synthetic data technically follow a highly positively skewed distribution, due to the way they are created, but when plotted (e.g., using around 20 or so histogram bins), the shape closely resembles an exponential distribution. For this reason, we used the more general term 'skewed' throughout the manuscript. We intend to keep this term, but add a clarifying sentence before **Line 206**, to explain that the distribution in practice appears similar to an exponential:

add>> "***Although the data created this way follow a highly positively skewed distribution, their shape closely resembles an exponential distribution when visualized with an insufficient number of histogram bins, due to the generation process.***"

Regarding introducing a reason for also using skewed distributions, we have added the following sentence in **Line 192**, after the first sentence of the paragraph:

add>> "***Nevertheless, numerous climate and other geoscientific model output variables do not follow a normal distribution, but instead exhibit skewed or exponential distributions, as is the case with daily precipitation (Ensor L. and Robeson S., 2008).***"

https://journals.ametsoc.org/view/journals/apme/47/9/2008jamc1757.1.xml

**Major Comment 7**

Line 207 I assume the iteration process is introduced due to instability introduced by the exponential transformation (which is a concern). If this is correct, why is iteration also applied to non-transformed data? Does that case also suffer from instability? Please clarify.

The iteration process was not introduced due to instability caused by the exponential transformation. Rather, it was applied in all cases because the metrics were computed using synthetic data generated through random sampling to match specific target statistics. Due to the stochastic nature of this process, individual realizations can exhibit variability. Repeating this process 200 times and using the median value ensures that the results are stable and robust. This repetition ensures convergence of the metric values and minimizes sensitivity to random fluctuations.

We now clarify this part of the text:

old>> "*For each combination of the aforementioned parameters, the procedure was repeated 200 times, and the median of the values of each metric was used.*"

new>> "*For each combination of the aforementioned parameters, and both the normally and skewed distributed cases, the procedure was repeated 200 times, and the median of the values of each metric was used, to ensure convergence of the metric values.*"

**Major Comment 8**

Line 208, 235 It appears that the "modifications" refer to small adjustments (e.g., subtracting 1), but I believe these should be explicitly stated or shown in an appendix or supplementary material.

We agree with the reviewer. We will now include Appendix, showing the exact form of each of the modified metrics.

add>> "*Appendix A: Modification of Metrics*

*The SPAEF, E , and MSPAEF metrics were modified to take values equal to or greater than zero, for the inter-comparison of the metrics*

*Specifically, the SPAEF metric was modified as follows:*

*SP AEF = sqrt((α − 1)2 + (β − 1)2 + (γ − 1)2)*

*The Esp metric was modified as follows:*

*Esp = sqrt((rs − 1)2 + (γ − 1)2 + (α − 1)2)*

*The MSPAF metric was modified as follows: M SP AEF = sqrt((α − 1)2 + (β)2 + (γ)2 + (δ)2)*"

**Major Comment 9**

Line 212 The explanation is clear, but the term "λ-δ plot" is introduced as if it were standard, which may not be the case. Also, since the plots involve correlation (ρ), the name might more appropriately reflect all components (e.g. λ-δ-ρ). Additionally, Figures 2–5 are referenced before Figure 1, which disrupts the reading flow. Consider expanding Figure 1 to include all components ($\lambda$, $\delta$, $\rho$) and use it as a comprehensive illustrative example.

Thank you for the feedback. We have created an illustrative example plot that looks like Figs 2-5 (see Figure 2). We also replace the paragraphs starting in **Lines 211 and 217** with the following:

old>> "*In these adjusted conditions, where zero indicates a perfect match with the observations, a well-behaved metric is expected to show decreasing values as the correlation coefficient increases and the bias decreases. In the $\lambda$-$\delta$ plots, this is reflected as the curves shifting at lower coordinates values as one moves towards the right and upward parts of the subplots (e.g., in Figures 2 to 5). Additionally, the lowest metric value for any combination of correlation and bias is anticipated at a standard deviation ratio of 1. This is reflected in the curve minimum being at a standard deviation ratio of 1 for each subplot, with metric values increasing as the standard deviation ratio deviates from 1.*

[Figure]

*Old Figure 1: Examples of metric behaviour curves for a given value of correlation and bias. Black curves indicate a well-behaved metric, while the blue curves indicate a poorly behaving metric. In the figure, $\lambda$ is the standard deviation ratio target between the model and the observations dataset*

*In the examples of Fig. 1, the black curves indicate well-behaving metrics, since for all of them, the minimum values are found at $\lambda$ of 1, and the values increase monotonically as the standard deviation ratio deviates from 1, even though some of the curves are not perfectly symmetric. The blue curves indicate poorly behaving metrics because, for curve e, the minimum value is not found at a standard deviation ratio of 1, while for curve d, the curve does not monotonically increase to the left side of the minimum.*"

new>> "*In these adjusted conditions, where zero indicates a perfect match with the observations, a well-behaved metric is expected to show decreasing values as the*

*correlation coefficient increases and the bias decreases. In the λ-δ-ρ plots, this is reflected as the curves shift at lower coordinates values as one moves towards the right and upward parts of the subplots, as illustrated by the purple curves in Fig. 2. Additionally, the lowest metric value for any combination of correlation and bias is anticipated at a standard deviation ratio of 1. This is reflected in the curve minimum being at a standard deviation ratio of 1 for each subplot, with metric values increasing as the standard deviation ratio deviates from 1.*

[Figure]

*Figure 2: Examples of metric behaviour curves for a given value of correlation and bias. Black curves indicate a well-behaved metric, while the blue curves indicate a poorly behaving metric. In the figure, $\lambda$ is the standard deviation ratio target between the model and the observations dataset.*

*In the examples of the top left subplot of Fig. 2, the black and purple curves indicate well-behaving metrics, since for all of them, the minimum values are found at λ of 1, and the values increase monotonically as the standard deviation ratio deviates from 1, even though some of the curves are not perfectly symmetric. The blue curves indicate poorly behaving metrics because, for curve e, the minimum value is not found at a standard deviation ratio of 1, while for curve d, the curve does not monotonically increase to the left side of the minimum."*

**Major Comment 10**
Line 239 It would be useful to specify the range of each statistic used in the comparison.

Thank you for the comment. We have added the following sentence to specify the range of each statistic used:

add>> *"The correlation coefficient, bias, and standard deviation ratio were each sampled at discrete intervals within the following ranges: correlation (−0.9 to 0.9), bias (0 to 3), and standard deviation ratio (0.3 to 1.8)."*

**Major Comment 11**

Section 3.2 This section introduces an interesting exercise, but lacks key contextual information. Please describe the motivation and objectives of the parameter variations (line 291). What are you trying to illustrate by modifying these parameters? How are the parameters changed and which are the hyperparameters? How did you select these hyperparameters? For instance, if a uniform distribution U(a = −1, b = 1) is used to model bias, why was this range chosen?

Additionally, consider whether the comparison is fair across models (A and B) and variables (precipitation and temperature). For example, is it common for precipitation model outputs to exhibit negative spatial correlation with observations? Why are correlation values so similar for temperature models (A and B)? Are the data transformed for precipitation, considering it typically does not follow a normal distribution? Given the applied nature of your work, an exploratory data analysis would help support the assumptions and setup. For instance, if temperature models replicate the mean very accurately, the insensitivity of SPAEF to bias may not be a serious issue. Presenting an extreme bias scenario (e.g., 7.5 Kelvin) may be less meaningful unless the goal is to show theoretical failures of other methods, rather than plausible real-world behavior. A justification for the selection of the specific scenarios you are presenting must be included.

In this part of the analysis, the input parameters (bias, standard deviation ratio, and correlation) were not varied across the 20 runs shown in Fig. 6b. Instead, the same parameter values were used to generate 20 independent realizations of synthetic data, due to the stochastic nature of the random sampling process. The use of boxplots allows us to show the spread and median of the metric values across these realizations, helping reduce the effect of the random variation in the data. We revised the text to clarify this and avoid the misleading use of the term "variations.":

old>> *"In Fig. 6b, the values of the four metrics are shown for 20 different variations with the aforementioned parameters."*

new>> *"In Fig. 7b, the boxplots show the distribution of the values of the four metrics from 20 different realizations of the synthetic data generated with the aforementioned parameters."*

**Line 306** will also be similarly revised.

old>> *"Figure 7b presents the values of the four metrics, for 20 different variations with the same values of parameters."*

new>> *"In Fig. 8b, the boxplots show the distribution of the values of the four metrics from 20 different realizations of the synthetic data generated with the aforementioned parameters."*

The goal of this section was to illustrate the behavior of the metrics under controlled synthetic conditions and highlight their sensitivity to somewhat atypical but relatively realistic cases of bias and spatial correlation.

It is common for simulated precipitation to exhibit negative correlation compared to observations in the meteorological sense (comparing short time frames). Similarly, in climatological timescales (e.g., annual averages) it would also not be unusual for it to have negative correlation regionally, since climate models are known to have large precipitation biases, especially near the equator, due to incorrect representation of the location of main atmospheric circulation features such as the Inter-Tropical Convergence Zone (ITCZ). Nevertheless, it would indeed be pretty unusual when taking into account the whole planet.

The correlation values are quite similar between the temperature models, to better illustrate the overwhelming effect of the mean bias, against small differences in the spatial pattern matching, in the different metrics.

The data were not transformed to a skewed distribution for the precipitation example. Instead, a normal distribution was used as in the temperature case. While precipitation is typically a skewed distribution, this is not always the case. For example, as seen in the third figure of the attached material, annual precipitation can exhibit a bimodal pattern, reflecting a combination of both skewed and slightly skewed distributions. If the polar regions (all areas beyond 75 degrees latitude at either hemisphere) are excluded, in order to focus in the tropical and extra-tropical regions, the resulting distribution of annual precipitation is only slightly positively-skewed and can be reasonably approximated by a normal distribution. On the other hand, daily precipitation, seems to generally follow an exponential distribution (or a highly positively skewed distribution).

Additionally, although these extreme bias examples for the temperature variable might look unrealistic in a global sense, in a local environment (e.g. country level) such large biases might occur momentarily.

**Major Comment 12**
Line 316 When you state, "we averaged over the 1981–2010 period," do you mean that each grid cell represents the average over all years for that location? If so, is this a standard approach for model evaluation? This process may result in considerable loss of information, so a reference would be helpful.

We would like to clarify that each grid cell represents the average annual value of the variable over the specified period for that specific location. We agree that this averaging may lead to a significant loss of temporal variability information. Nevertheless, in climatology, it is a standard practice to evaluate model performance and spatial patterns based on temporally averaged fields over several decades.

We could add the following sentence at **Line 316**:
 add>> "*These multi-decadal averages were used to reduce the short term variability, and highlight the long term climatological signal (Nooni I. Et al, 2022 ;Du Y. et al, 2023).*"

https://rmets.onlinelibrary.wiley.com/doi/abs/10.1002/joc.7616
https://www.mdpi.com/2073-4433/14/3/607

**Major Comment 13**

Section 3.3 Following up on comments regarding Section 3.2, I'm concerned about the similarity among bias-insensitive metrics and MSPAEF. How are you defining "similarity" between model output and observations? The issue resembles your first synthetic example. I believe that whether a metric should favor spatial pattern accuracy over mean accuracy or vice versa may depend on the application. If your primary aim is to detect spatial similarity, that should be explicitly stated, but for now this is just inferred.

Consider also that some variants of the Kling-Gupta Efficiency (KGE) allow weighting of each component, which also may be useful for tuning "similarity" in your selected "metrics". However, this requires the user to define those weights. As an alternative, your exploratory analysis (from Section 3.2) could guide which metric components need stronger discrimination. With these variations, some methods may perform comparably to MSPAEF, without diminishing the merit of the interesting properties of this new indicator. This is something that you should try or at least mention.

The similarity of MSPAEF to the bias-insensitive metrics might look a bit strange at first glance (at least compared to the other bias-sensitive metrics), since this is a bias-sensitive metric. Nonetheless, this is not as odd when we take into account the large sensitivity of WSPAEF (the other bias-sensitive metric) to the absolute value of the mean bias. This sensitivity is what causes its performance to diverge significantly from that of MSPAEF (and the other two bias-insensitive metrics).

We also agree that whether a metric should prioritize spatial pattern accuracy or mean bias accuracy depends on the application. We defined "similarity" between model output and observations as a combination of both spatial pattern agreement and agreement in the mean values. The MSPAEF metric was designed as a comprehensive similarity measure, that responds to both spatial pattern agreement and bias in the mean. It is defined to emphasize spatial pattern similarity when the relative bias is small, but to increasingly emphasize the bias as it becomes more significant. This design allows MSPAEF to act as a balanced indicator, adapting its emphasis depending on the characteristics of the data.

We appreciate your recommendation regarding the potential use of weighted variants of existing similarity indices. While MSPAEF was developed as a fixed form, non-tunable metric to maintain consistency across different contexts, we agree that incorporating adjustable weighted schemes could allow for shifting the emphasis on pattern or bias, depending on the application at hand.

In **Sect. 3.3**, we now mention which components will need to have a greater contribution in order for the existing measures to perform similarly to MSPAEF:

add>> "*Using weights for the different components could serve as a way to improve the performance of the existing metrics, to closely match the performance of MSPAEF. For example, in the case of WSPAEF, achieving a behavior more consistent with MSPAEF in the presence of significant absolute mean bias would require reducing the relative contribution of the WD component. By contrast, SPAEF and E_{sp} already perform similarly to MSPAEF when bias values are small, but as bias increases, their lack of bias-sensitive components will limit their ability to achieve similar performance to MSPAEF.*"

**Major Comment 14**

Table 1, Table 2 Sometimes, it is necessary to rank models based on two or more variables simultaneously. You are currently ranking the models using only one variable at a time. Could you consider adding references that illustrate methods for multi-criteria model ranking?

In this study, the CMIP6 models were ranked separately for each variable and each metric, resulting in independent rankings that highlight how the models' performance varies with different variables. we agree that in many practical applications, model performance must be judged using multiple variables simultaneously. We have added the following paragraph:

add>> "*While in this demonstration we evaluate and rank models separately for each variable and each metric, in many real-world applications, the overall model performance can be assessed using multiple variables. There are many multi-criterion model ranking techniques that can do this, such as Compromise Programming (CP) (Refaey M. A. et al., 2019; Baghel T. et al., 2022), Technique for Order Preference by Similarity to an Ideal Solution (TOPSIS) (Srinivasa Raju K. and Nagesh Kumar D., 2014) and Cooperative Game Theory (Gershon M. and Duckstein L., 1983).*"

https://www.ajbasweb.com/old/ajbas/2019/May/85-96(9).pdf
https://www.sciencedirect.com/science/article/abs/pii/S0048969722016448
https://iwaponline.com/jwcc/article-abstract/6/2/288/1601/Ranking-general-circulation-models-for-India-using?redirectedFrom=fulltext
https://ascelibrary.org/doi/10.1061/%28ASCE%290733-9496%281983%29109%3A1%2813%29

**Anonymous Referee #1 (RC2)**

Most of my comments were adequately addressed or corrected. However, I still have a few remaining observations. I will continue using the original numbering from my first round of comments.

We're very grateful for the thoughtful time and effort you put into giving us constructive feedback.

**Major Comment 7**

Following your explanation, I now understand that (even after adjusting for bias, standard deviation, and correlation between observed and modeled data) there can still be non-trivial spatial variations that affect the behavior of certain "metrics". If this interpretation is correct, I recommend stating it explicitly in the manuscript. This clarification would also apply to the evaluation of synthetic data in special-case scenarios.

We would like to further clarify that even when synthetic datasets are generated with specific target values for correlation, standard deviation ratio, and bias, due to the stochastic nature of the process, the exact targets are not always precisely achieved. By repeating the generation process multiple times and using the median value of the metrics that are applied in these synthetic data, we reduce the impact of outliers and approximate the behavior of the metrics under the intended statistical conditions.

We agree with the referee that, beyond minor deviations from the target statistics, there can also be non-trivial spatial variations in the synthetic data fields. These variations can influence the performance of different metrics in distinct ways. We will revise the manuscript to state this explicitly, as this is particularly relevant for interpreting the metrics responses in synthetic-data experiments, including special-case scenarios.

We will revise this part further:

old>> "*For each combination of the aforementioned parameters, the procedure was repeated 200 times, and the median of the values of each metric was used.*"

new>> "*For each combination of the aforementioned parameters, and for both the normally and skewed distributed cases, the procedure was repeated 200 times, and the median value of each metric was used to ensure convergence of the metrics. While the synthetic data are generated to match specific target values of correlation, bias, and standard deviation ratio between them, the stochastic nature of the process means these targets are only approximately achieved. Additionally, non-trivial spatial variations may still occur across realizations, which can affect different metrics in distinct ways. The repetition and the use of the median metric values help reduce the influence of these variations and provide a more robust estimate of each metric's behavior under the intended conditions.*"

**Major Comment 11**
Regarding the concerns I previously raised (e.g., negative spatial correlation, use of normally distributed data for precipitation), your explanation is satisfactory. However, I still believe it is important to include these justifications in the manuscript. Doing so would help support your choice of parameter values and make the rationale clearer to readers. Please include references.

We will modify **Line 278** as follows to clarify that these examples use normally distributed synthetic data:

old>> "*Two examples are presented to illustrate the differences in the values and the interpretation of the four metrics, using synthetic data as described in the Methods section.*"

new>> "*Two examples are presented to illustrate the differences in the values and the interpretation of the four metrics, using normally distributed synthetic data as described in the Methods section.*"

To address the use of negative correlation in Example 1, we will add the following at **Line 285**:

add>> "*While the use of negative spatial correlations might look unusual, especially at global scales, they can occur regionally, especially in precipitation fields due to known large biases near the equator, such as the double ITCZ bias (Ma X. et al., 2023).*"

For the choice of normal distribution for the precipitation example, we will add the following at the end of **Line 283**:

add>> "*Although daily precipitation often follows a highly skewed or exponential distribution, annual averages can be closely approximated to a normal distribution, especially outside of polar regions (see figure in Appendix).*"

Regarding the use of extreme bias in Example 2, we will add the following at the end of **Line 300**:

add>> "*While this large bias example might seem unrealistic at global scales, it can occur at local scales and/or momentarily (McSweeney C. F. et al., 2015).*"

https://rmets.onlinelibrary.wiley.com/doi/abs/10.1002/joc.7980
https://link.springer.com/article/10.1007/s00382-014-2418-8

**Major Comment 13**

Again, your explanation is clear and satisfactory (particularly regarding why MSPAEF appears similar to other bias-insensitive "metrics". I suggest incorporating this discussion into the manuscript as well. Additionally, it would strengthen the paper to reflect some of these insights in the **Conclusions** section. For instance, you could highlight the idea that certain "metrics" may be improved by incorporating weights, while in other contexts, their unweighted forms may be sufficient.

I also appreciate that you now mention some limitations of the MSPAEF. However, I recommend improving the explanation of its advantages. Although I am aware of the existence of this explanation in the manuscript, I found the following paragraph stronger and suggest including it with the necessary changes:

"We defined 'similarity' between model output and observations as a combination of both spatial pattern agreement and agreement in the mean values. The MSPAEF metric was designed as a comprehensive similarity measure that responds to both spatial pattern agreement and bias in the mean. It is defined to emphasize spatial pattern similarity when the relative bias is small, but to increasingly emphasize the bias as it becomes more significant. This design allows MSPAEF to act as a balanced indicator, adapting its emphasis depending on the characteristics of the data."

We are grateful for your constructive suggestions. For the similarity of MSPAEF to the bias-insensitive metrics, we will add the following paragraph after **Line 357**:

add>> "*The similarity of MSPAEF to the bias-insensitive metrics might look unusual at first glance (compared to the other bias-sensitive metrics), given that MSPAEF is a bias-sensitive metric. However, this is less surprising when considering the large sensitivity of WSPAEF (the other bias-sensitive metric) to the absolute value of the mean bias. This sensitivity is what causes WSPAEF's performance to diverge significantly from that of MSPAEF (and the other two bias-insensitive metrics), particularly for the precipitation variable, where the bias in the mean can be large in absolute terms.*"

We will add the following paragraph after **Line 387**, to reflect some insights in the **Conclusions section**:

add>> "*Through the use of appropriate weights, the existing metrics can be adjusted to better align with the performance of MSPAEF. Nevertheless, on many occasions, such as when there is insignificant bias, their original unweighted forms often perform sufficiently well, reducing the need for such modifications.*"

We will revise the **last paragraph of the conclusions section**, to emphasize the advantages of MSPAEF:

old>> "*Although this metric was developed for evaluating gridded climate model output, its underlying rationale and its inherent flexibility make it suitable for assessing the performance of other types of geoscientific or environmental models where the spatial distribution of simulated variables is expected to follow certain patterns in space.*"

new>> "*The MSPAEF metric was designed as a comprehensive similarity measure that incorporates both spatial pattern agreement and bias in the mean. It is defined in a way that emphasizes spatial pattern similarity when the relative bias is small, but it increasingly emphasizes the bias as it becomes more significant. This design allows MSPAEF to act as a balanced indicator, adapting its emphasis depending on the characteristics of the data. While this metric was developed for evaluating gridded climate model output, its underlying rationale and its inherent flexibility make it suitable for assessing the performance of other types of geoscientific or environmental models where the spatial distribution of simulated variables is expected to follow certain patterns in space.*"
* * *
**Anonymous Referee #2 (RC3)**

General Comments

This paper is structured well and describes a new metric for spatial correspondence in evaluation of climate models with observations. However, the novelty factor is not very clear in this work as it largely draws on previously established metrics.

The MSPAEF metric takes into account bias in its calculation through the NRMSE primarily so it is to be expected that will be reflected in the results when compared to other metrics that explicitly don't. Two key questions arise:

**General Comment:**

1. It is generally accepted in the climate science community that multiple metrics should be taken together to assess model performace in evauation. Some of this may also be region/variable dependent. What is the justification for not simply using the SPAEF or Esp metrics and looking at bias as well simply through the NRMSE score and perhaps other measures as well? (For e.g. see Gomez et al in https://doi.org/10.1002/joc.8619). In fact the WSPAEF does well for their choice of region it seems.

We appreciate this important point. We agree that for a comprehensive climate model evaluation, one should rely on multiple complementary metrics, and that MSPAEF is not intended to replace existing, simpler measures like NRMSE, or more specialized metrics such as SPAEF or Esp. Rather, the motivation behind MSPAEF is to provide a single, balanced indicator that combines two key aspects of model performance—spatial pattern similarity and mean relative bias—in a unified framework.

An important distinction of the MSPAEF, compared to other similar metrics, is its adaptivity. Unlike purely spatial metrics, MSPAEF automatically shifts weight between bias and spatial pattern similarity depending on the magnitude of the relative bias. When the bias is small, the metric behaves very similarly to SPAEF or Esp; when the bias is large, it appropriately penalizes the model. This adaptive behavior provides an advantage in contexts where the relative importance of bias is also relevant.

We fully agree with the reviewer that multi-metric approaches remain essential. MSPAEF here is proposed as an additional option. One that can be particularly helpful when a single summary statistic is desired, while still respecting both spatial and mean characteristics. Using a single bias-sensitive metric can also be advantageous because it avoids the ambiguity that may arise when separate spatial and bias metrics are in disagreement. By combining these two aspects within a unified framework, MSPAEF provides a clearer and more consistent basis for ranking or comparing models. In this sense, MSPAEF fills a gap between purely spatial metrics and purely bias-focused ones.

Furthermore, we acknowledge that WSPAEF can perform well for certain regions, as noted by the reviewer and also consistent with Gomez et al. (2024). Part of our motivation for MSPAEF is precisely that WSPAEF's performance can vary substantially with the absolute magnitude of the bias, and therefore with the choice of variable and units, which may lead to inconsistencies if not applied with caution. MSPAEF was developed to provide a more stable and balanced behavior across a wider range of conditions.

Taking into account these aspects, in the revised version we will add the following paragraph at the end of the introduction:

add>> "*It is important to stress that the MSPAEF metric is not intended to replace traditional multi-metric evaluation approaches. Instead, it is designed to complement existing measures by providing a balanced indicator that captures both spatial pattern similarity and relative mean bias within a single metric. In this way, MSPAEF serves as a useful additional tool, particularly when a unified summary statistic is desirable.*"

We will also modify the last paragraph of the conclusions:

old>> "*Although this metric was developed for evaluating gridded climate model output, its underlying rationale and its inherent flexibility make it suitable for assessing the performance of other types of geoscientific or environmental models where the spatial distribution of simulated variables is expected to follow certain patterns in space.*"

new>> "*The MSPAEF metric is not proposed as a substitute for multi-metric evaluation, which remains essential in climate model assessment. It was designed as a comprehensive similarity measure that incorporates both spatial pattern agreement and bias. It is defined in a way that emphasizes spatial pattern similarity when the relative bias is small, but it increasingly emphasizes the bias as it becomes more significant. This design allows MSPAEF to act as a balanced indicator, adapting its emphasis depending on the characteristics of the data. While this metric was developed for evaluating gridded climate model output, its underlying rationale and its inherent flexibility make it suitable for assessing the performance of other types of geoscientific or environmental models where the spatial distribution of simulated variables is expected to follow certain patterns in space. Its use alongside established measures can help provide a clearer and more consistent picture of model performance.*"

2. Spatial correlations have been used to identify model groupings/famllies (https://doi.org/10.1198/016214507000001265). It seems like that might not be so obvious from Figure 8 in the paper (the MIROC model stands out as an example, CNRM is another where the higher resolution model seems to do better. This is perhaps expected but NorESM2-LM is very far apart from #5,#6 and #32 which seems odd.). Can the authors offer some hypotheses for this?

We thank the reviewer for raising this interesting point. We agree that previous studies, such as the one cited, have shown that climate models developed by the same institution or sharing a common parent model often exhibit similar spatial error structures and therefore tend to cluster when evaluated using spatial correlation–based diagnostics. In our results (Fig. 8), we observe that some model families indeed cluster closely, but others show a much wider spread.

One of the striking differences is that of NorESM2-LM, which does not cluster with the rest of the group. For that one, we see that the differences are found just for the precipitation variable. The NorESM2-LM is run at a 2-degree resolution for the atmospheric component, while the NorESM2-MM and the CESM2s are run at 1 degree resolution. This large difference in the resolution can have a large impact on the representation of the local precipitation patterns, which are more likely to lead to larger local biases and therefore significantly poorer overall model performance.

Another example is the MIROC-ES2L and MIROC6 models. In Fig. 8, it is obvious that MIROC-ES2L performs much worse than MIROC6, for both variables. MIROC-ES2L native atmospheric resolution is about 5 degrees, while for MIROC6, it is about 2.5 degrees. This large difference in the resolution can, for the same reason as above, explain the differences in the precipitation performance. We also note that MIROC-ES2L is by far the worst-performing CMIP6 model for the temperature. Its coarse grid blends land and ocean surfaces, especially near coastlines, which can significantly distort regional temperature fields. It is therefore expected that MIROC-ES2L would perform worse than a higher-resolution version of the same model family, which can resolve coastlines and regional gradients more accurately.

Generally, we observe that the outliers in the performance from models of the same model family or institution are the ones that have significantly coarser atmospheric resolution. Lower resolution naturally alters the representation of key processes and leads to markedly different precipitation patterns and therefore affects the overall model performance. Other factors, such as different parametrizations and internal variability of the models can also have some effect on the differing model performances, but the systematic link between resolution and performance provides the most plausible explanation for this discrepancy.

To discuss this issue, we will add the following paragraph in **section 3.3**, before **Line 325**:

add>> "*In line with a previous study (Jun et al., 2008), we generally expect models from the same developers or model family to cluster together in performance, and we do observe this behavior for most groups. However, a few clear outliers emerge. These outliers typically correspond to models with substantially coarser atmospheric resolution (see Appendix C), which can markedly alter precipitation characteristics and therefore degrade overall performance relative to their higher-resolution counterparts. Although differences in parameterizations and internal variability may also play a role, the systematic link between resolution and performance offers the most plausible explanation for the departures from the expected clustering.*"

To facilitate the interpretation of results, we will also add a table in the appendix (**Appendix C**) with details about each of the CMIP6 models used (Model ID, Model name, Parent Organization, Atmospheric resolution, Ocean Resolution, and references).

**Specific Comments:**

1. A question about the synthetic Model data used in Section 3.2 is whether this is realistic. The grid cells are uniform and area averaging is not needed which is not the case when

comparing model and observations traditionally. It is not clear if the bias values in the models would be realistic under those circumstances.

The primary objective of Section 3.2 is to isolate and illustrate the theoretical behavior of the MSPAEF metric. For this reason, we intentionally constructed a simplified synthetic experiment on a uniform grid without applying area-weighting. This controlled setup allows us to vary mean bias and spatial pattern errors independently, which would not be possible in a realistic, irregular grid. The purpose here is therefore a conceptual demonstration of the different responses of the four metrics. In practical applications, especially over large or global domains, MSPAEF can be combined with area-weighting if desired, though the metric itself should give very reasonable results even without it. Over limited regions, like continents, where the grid-cell areas are more homogeneous, the bias values and behaviors shown in our synthetic experiment are expected to be very representative.

2. There is no discussion of how differences in model and observation resolution will be accounted for. Regridding will impact many terms in the calculation of MSPAEF. Would the inferences made regarding the relative performance still hold for the MSPAEF and why?

Thank you for this comment. We agree that differences in the model and observational resolution must be handled carefully when applying any spatial metric, including MSPAEF. In practice, MSPAEF is computed after the two datasets have been interpolated into a common grid, following the standard practice in model evaluation. This regridding might indeed have an effect on all spatial metrics that rely on grid-point values. However, the relative performance rankings among models are expected to remain largely consistent because all models have undergone the same regridding procedure to the same target grid before MSPAEF is computed. In this way, any smoothing or distortion that occurs due to the regridding, is expected to have a similar effect across the datasets. We will clarify in the manuscript that comparisons should always be done on a common grid, and that MSPAEF behaves consistently under such standard preprocessing.

In addition to our original approach, we tested regridding only the observational data at the resolution of the CMIP6 model output, and we see very similar results, as when regridding everything to 1 degree resolution (see attached figure Tas_vs_pr_CMIP6_new.pdf)

To avoid any confusion, we will add the following paragraph at the end of **section 2.5** (Real climate data):

add>> "*To compute MSPAEF or any other spatial metrics, the model and observational datasets need to be placed into a common grid. Regridding data can generally affect the values of metrics that rely on grid-point values. However, the relative performance rankings among models are expected to remain largely unchanged because all models undergo the same regridding procedure to the same target grid. Therefore, any smoothing or distortion that occurs due to the interpolation is expected to have a similar and small effect across the datasets, provided that the regridding does not involve extreme changes to resolution.*"

3. A key component of model evaluation is looking at the spatial distribution of values. In lines 315-325, the authors mention how some models perfom better or worse in spatial distribution comparisons but it is not clear how that is reflected in Figure 8. For instance,

MIROC6 shows a score of ~0.84 for precipitation and ~0.91 for temperature.but is said to perform badly for spatial distribution of temperature. However, it is not quite clear from Figure 8, what the difference between MROC6 and #15,  EC-Earth3 (slightly better) is when compared to say  #29, MRI-ESM2-0. In other words, what do the numbers really signify on a map and how does a number for temperature compare for a number for precipitation for instance. It would be very useful to unpick this information for the metric to be truly useful.

Thank you for this observation. Understanding what a given value of the MSAPEF means on a map is essential for perceiving the model performance. Since MSPAEF is a compound metric that combines spatial and bias-focused measures, these two characteristics can both affect the final value of the metric. Specifically, it is affected by how well the patterns of the variable match between the two datasets (in maps is seen by how well the location and intensity of the gradients of the variable match), the relative mean bias difference, and the differences at each grid-point (which is affected by both changes in the location and intensity of the gradients in a map, but also the mean bias).

In the case of MIROC6, for instance, the relatively high value of MSPAEF for temperature reflects a good overall performance, but it masks the fact that this model exhibits comparatively weaker performance in recreating the spatial patterns and values of the temperature, compared to most other CMIP6 models. On the other hand, the significantly higher value of MSPAEF for temperature for the MRI-ESM2-0, indicates that this model better represents not only the spatial patterns, but also the actual values of temperature at each grid-point.

Comparing MSPAEF values for different variables is a bit trickier. In the example shown in the manuscript, we generally see that the models have higher values of the MSPAEF metric for the temperature compared to precipitation. This generally indicates that the spatial pattern of temperature, as well as the grid-point values, are more easily reproduced in the models, due to their smoother nature, at least compared to precipitation, where steeper gradients are more common due to the underlying terrain, and differences in the large-scale circulation. In general, the higher values of MSPAEF for temperature indicate that the models can better reproduce the underlying patterns of this variable, and at the same time, the values at the grid-point level differ less when taking into account the variability of the whole domain.

In the revision, we will modify and split the paragraph at lines 325-332 to better reflect the meaning of these values:

old>> "*Regarding the actual values of the metric, all CMIP6 models except MIROC-ES2L, have values greater than 0.9 for the temperature variable. Contrariwise, all models except the two CESM2s and the NorESM2-MM models have values less than 0.85 for the precipitation variable. This indicates that the CMIP6 models generally can capture well the spatial distribution and magnitude of the temperature, while they struggle a lot more in the representation of the precipitation features. This is somewhat expected, as the spatial distribution of temperature is influenced by factors such as orography, latitude, and proximity to the oceans, which are easier to be represented in climate models. Modeling precipitation, on the other hand, is significantly more complex (Legates, 2014; Räisänen, 2007). It is largely influenced by sub-grid-scale processes like convection and cloud microphysics that are not explicitly resolved in the models, and they need to be parametrized (Pieri et al., 2015).*"

new>> "*Regarding the actual values of the metric, all CMIP6 models except MIROC-ES2L, have values greater than 0.9 for the temperature variable. MRI-ESM2-0 is the model with the best performance for the temperature variable, which indicates that this model better represents not only the spatial patterns, but also the actual values of temperature at each grid-point. Contrariwise, all models except the two CESM2s and the NorESM2-MM models have values less than 0.85 for the precipitation variable. This indicates that the CMIP6 models generally can capture well the spatial distribution and magnitude of the temperature, while they struggle a lot more in the representation of the precipitation features.*

*Although comparing the MSPAEF values for two different variables is not straightforward, some conclusions on the nature of the variables can be deduced. The spatial distribution of temperature is influenced by factors such as orography, latitude, and proximity to the oceans, which are easier to be represented in climate models, due to their generally smoother and slow-varying nature in space. On the other hand, precipitation tends to have steeper gradients. Modeling precipitation is significantly more complex (Legates, 2014; Räisänen, 2007), as it is largely influenced by sub-grid-scale processes like convection and cloud microphysics that are not explicitly resolved in the models, and they need to be parametrized (Pieri et al., 2015), which makes it more difficult for climate models to correctly capture the underlying gradients and the actual grid-point values.*"

Moreover, we will add an extra paragraph in the subsection where MSPAEF is defined, to signify what exactly the output numbers of MSPAEF signify on a map (last paragraph on the **Response to Technical Comment 1**).

**Technical Comments:**
1. It would be good to expand the MSPAEF in the subsection where the metric is defined

Following the reviewer's feedback, we will expand the MPSPAEF definition by adding the following paragraphs at the end of the subsection:

add>> "*A further advantage of normalizing all components by IQR and expressing each term in dimensionless form is that MSPAEF becomes relatively insensitive to preprocessing decisions such as unit conversions, domain rescaling, or masking of small regions. Unlike WSPAEF, whose values are affected by the unit of choice, MSPAEF can produce comparable values under a wide range of analysis settings, which increases reproducibility across studies and domains.*

*An important motivation for this definition of MSPAEF is to penalize inconsistent model behavior, where a model with poor spatial correlation but a very small bias (or vice versa) can appear artificially good in composite metrics. Since MSPAEF treats each discrepancy in either the spatial patterns or in the mean relative bias as an orthogonal dimension, it prevents a strong performance in one characteristic from masking deficiencies in the other. This provides a more complete evaluation of how each model differs from observations.*

*Based on its definition, MSPAEF is sensitive to both spatial pattern agreement and mean bias, and this sensitivity can be directly interpreted in the corresponding spatial maps. Each component of the metric can be associated with an observable feature in the maps. The spatial correlation term represents how well the locations and intensities of spatial gradients align between the model and observations. The NRMSE identifies differences in grid-point magnitudes across the domain that are linked to both the location and intensity of gradients, but also to the mean bias. The relative mean bias term quantifies systematic offsets in the average values of the field. Finally, the variability term reflects how well the model reproduces the amplitude of spatial fluctuations. Together, these components allow changes in the MSPAEF value to be directly linked with spatial mismatches.*"

We will also add the following references at **Line 149**:

old>> "*The normalized RMSE (β term) is computed using the interquartile range (IQR) of the observations rather than the standard deviation, as IQR provides a more robust measure of variability*"

new>> "*The normalized RMSE (β term) is computed using the interquartile range (IQR) of the observations rather than the standard deviation, as IQR provides a more robust measure of variability (Rousseeuw P., Hubert M., 2011; Huber P., Ronchetti E., 2009)*"

2. Figure 1 is not mentioned anywhere in the text.

Old Figure 1 will be modified for greater clarity, following also the comments of Referee #1 (see Figure 1). More explanations will be added in the corresponding section, as seen in the Response to RC1 Major Comment 9.

3. Line 43: inhomogenous →nonhomogenous

Thank you. We will follow the reviewer's recommendation. The word has already been removed from a previous comment.

4. The y-axis in Figures 2-5 need labels.

Thank you for noticing this. We will add the labels on the y-axis of Figures 2-5.